# HiDe: Rethinking The Zoom-IN method in High Resolution MLLMs via Hierarchical Decoupling

## Abstract

Multimodal Large Language Models (MLLMs) have made significant strides in visual understanding tasks. However, their performance on high-resolution images remains suboptimal. While existing approaches often attribute this limitation to perceptual constraints and argue that MLLMs struggle to recognize small objects, leading them to use "zoom in" strategies for better detail, our analysis reveals a different cause: the main issue is not object size, but rather caused by complex background interference. We systematically analyze this "zoom in" operation through a series of decoupling experiments and propose the Hierarchical Decoupling Framework (HiDe), a training-free framework that uses Token-wise Attention Decoupling (TAD) to decouple the question tokens and identify the key information tokens, then leverages their attention weights to achieve precise alignment with the target visual regions. Subsequently, it employs Layout-Preserving Decoupling (LPD) to decouple these regions from the background and reconstructs a compact representation that preserves essential spatial layouts while eliminating background interference. HiDe sets a new SOTA on V*Bench, HRBench4K, and HRBench8K, boosting Qwen2.5-VL 7B and InternVL3 8B to SOTA (92.1% and 91.6% on V*Bench), even surpassing RL methods. After optimization, HiDe uses 75% less memory than the previous training-free approach. Code is provided in the supplementary.

## 1 Introduction

In recent years, Multimodal Large Language Models (MLLMs) have demonstrated remarkable capabilities in understanding and reasoning across diverse tasks such as image captioning, visual question answering, and image-text retrieval, emerging as a key driving force behind the advancement of multimodal artificial intelligence (Bai et al. (2023); Liu et al. (2023b); Bai et al. (2025); Zhu et al. (2025)). With continuous improvements in model architectures and the scaling up of training data, MLLMs have achieved significant progress in handling complex semantics and achieving fine-grained cross-modal alignment. However, recent studies (Zheng et al. (2025); Zhang et al. (2025a); Shen et al. (2024); Wu & Xie (2023); Wang et al. (2025); Li et al. (2025)) have revealed that existing approaches still fall short of expectations in handling high-resolution image tasks, as evidenced by their suboptimal performance on benchmarks such as V*Bench (Wu & Xie (2023)).

To enhance the model's perception of image details, existing methods often adopt a "locate-then-zoom-in" strategy. Training-based approaches, such as Supervised Fine-Tuning (SFT) (Shao et al. (2024)) or Reinforcement Learning (RL) (Zheng et al. (2025); Zhao et al. (2025)), can guide models to identify relevant regions. However, they suffer from critical drawbacks (Zhai et al. (2023); Yue et al. (2025)), including high costs, lengthy training, and a lack of cross-architecture transferability, which severely limits their scalability and practical deployment. Training-free methods (Zhang et al. (2025a); Shen et al. (2024); Li et al. (2025)), which automatically locate regions by analyzing attention or performing tree-based search, are often inefficient at inference due to multiple forward passes and exhibit high miss rates, especially when handling multiple objects, limiting their practicality.

However, we find that the widely adopted "zoom-in" strategy conflates upscaling and cropping and simply upscaling the image can not improve MLLM's preformance. To trace the true bottlenecks and

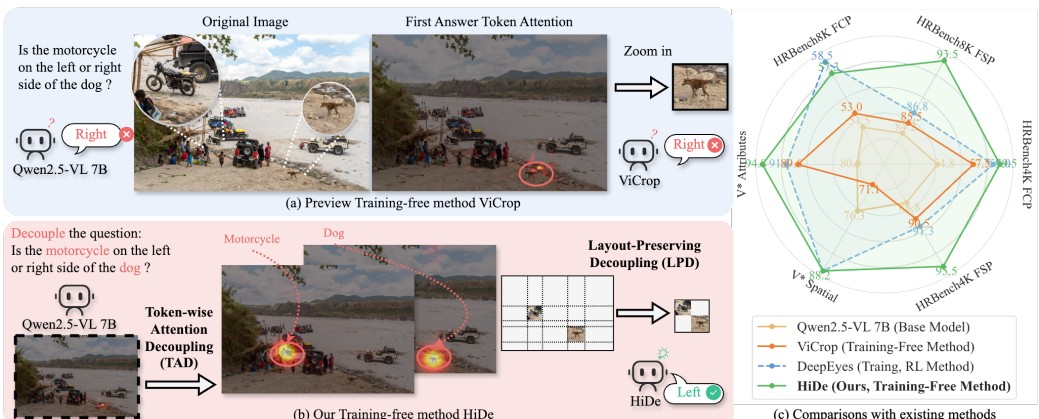

Figure 1: (a) Previous methods struggle to locate objects. (b) HiDe precisely locates objects and keeps relative positions. (c) HiDe outperforms previous training-free and beats the trained one.

isolate what actually improves performance, we conduct a **hierarchical decoupling analysis** (Fig. 2) starting from the zoom-in operation:

(i) **Zoom-in → upscale and crop**. Quantitative experiments find that scaling the whole image alone does not help MLLM decisions, indicating that cropping is the crucial component responsible for performance gains.

(ii) **Crop → foreground and background**. We separate the cropped region into foreground and background. We further decompose the **background** into **semantic distractors** and **token-level redundancy**, and validate that both factors concurrently introduce significant interference into the MLLM's reasoning process.

(iii) **Question text → semantic and non-semantic tokens**. We decouple the input text and verify that semantic tokens are the primary drivers of effective visual–text alignment and precise localization of target regions.

(iv) **Foreground → object appearance and spatial layout**. Leveraging objects identified by semantic tokens, We highlight that modeling the relative layout of objects is crucial to aid the model's judgment.

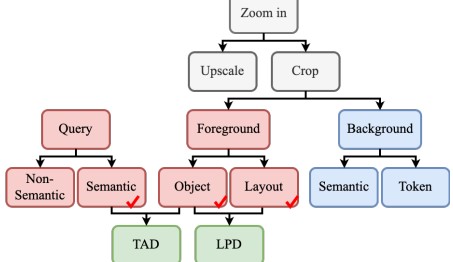

Figure 2: Hierarchical decoupling framework for analyzing MLLM performance on high-resolution images. The details in gray blocks are shown in Fig. 3. Blue Fig. 4. Red Fig. 5. Green Fig. 6

This structured decomposition reduces the problem to its minimal factors and allows us to selectively leverage the effective components while discarding the detrimental ones, thereby improving MLLMs' performance on high-resolution understanding.

Guided by these insights, we propose **HiDe**, a Hierarchical Decoupling method for precise and structured visual representation. As shown in Fig. 1, HiDe has two modules: **Token-wise Attention Decoupling (TAD)** cleans attention maps via token-level denoising; **Layout-Preserving Decoupling (LPD)** binarizes them to separate regions and preserves spatial layout via grid-based reconstruction.

We conduct extensive experiments on high-resolution datasets V* (Wu & Xie (2023)), HRBench-4K, and HRBench-8K (Wang et al. (2025)), with prominent MLLMs like Qwen2.5-VL and InternVL3. HiDe surpasses existing training-free methods and even outperforms RL-trained approaches on most tasks, achieving state-of-the-art results on both single-object and multi-object benchmarks. In addition, through a simple yet effective engineering modification, we reduce the peak memory usage from 96 GB to 20 GB (75%), substantially improving the practical applicability. Our contributions are summarized as follows:

- We perform a hierarchical decoupling of existing zoom-in approaches, revealing background distraction as the root cause of MLLMs' limitations on high-resolution images and isolating the components that genuinely drive performance gains.

- We propose HiDe, a hierarchical decoupling framework (TAD + LPD) that extracts precise, compact, and structurally coherent visual representation.

- Our method achieve state-of-the-art accuracy on both single- and multi-object tasks with only a small increase in inference overhead, and reduce memory footprint by 75% compared to previous training-free methods, enhancing the practicality of our method.

## 2 RELATED WORKS

**Multimodal Large Language Models (MLLMs).** MLLMS are foundational models that support diverse vision and language tasks. Early approaches typically use fixed-resolution vision encoders (e.g., 224×224 or 448×448) (Liu et al. (2023b); Bai et al. (2023); Li et al. (2023a; 2022); Liu et al. (2024; 2023a)), which simplify training but often lose fine details due to resizing or cropping—limiting performance in fine-grained tasks. To address this, Native/Dynamic-Resolution MLLMs have emerged, processing images at their original resolution and generating variable-length visual token sequences that adapt to input size. These models preserve spatial fidelity and support high-resolution inputs through mechanisms such as sliding window attention, dynamic masking, or patch-based encoding. Representative examples include InternVL3 (Zhu et al. (2025); Chen et al. (2024a); Wang et al. (2024b); Gao et al. (2024); Chen et al. (2024b;c)), which splits an image into fixed-size patches, encodes them separately with a standard ViT (Dosovitskiy et al. (2021)), and concatenates the resulting tokens; and Qwen2.5-VL (Bai et al. (2025); Wang et al. (2024a); Bai et al. (2023)), which trains a ViT end-to-end on native-resolution images to embed the full image into a single token sequence in one pass. Our work advances the understanding of how such models utilize visual information and proposes a scalable, training-free method to enhance their perceptual capabilities. This method offers orthogonal advantages to existing approaches, creating an effective complement.

**High-Resolution Visual Question Answering (HR-VQA).** HR-VQA (Wu & Xie (2023); Wang et al. (2025); Zhang et al. (2025b)) evolves the standard VQA paradigm (Li et al. (2023b); Kazemzadeh et al. (2014)) by shifting focus from holistic scene recognition to the precise perception of fine-grained details. However, MLLMs struggle with HR-VQA. To enhance capabilities of MLLMs in this area, research efforts have split into two directions, each with critical limitations. On one hand, training-based methods using supervised fine-tuning (SFT) (Shao et al. (2024); Wu & Xie (2023)) or reinforcement learning (RL) (Zheng et al. (2025); Zhao et al. (2025); Chen et al. (2025)) often lead to catastrophic forgetting (Zhai et al. (2023)) or sacrifice general robustness for task-specific gains (Yue et al. (2025)). On the other hand, training-free strategies (Zhang et al. (2025a)) that crop regions from attention maps or tree search strategies (Li et al. (2025); Wang et al. (2025)) are hampered by a confluence of issues: they are inefficient, incompatible with modern architectures like FlashAttention (Dao et al. (2022)) and prone to missing targets in multi-object scenes. While situated within the training-free paradigm, our work addresses the shortcomings of current methods by introducing a novel mechanism. This mechanism efficiently localizes multiple targets, offering a more flexible and powerful framework for complex, fine-grained visual tasks.

## 3 DECOUPLING MLLMS HIGH-RESOLUTION LIMITATIONS: ANALYSIS

In this section, we conduct a hierarchical decoupling analysis to identify the root causes of MLLMs' underperformance on high-resolution images and outline key insights for improving their effectiveness. The decoupling framework is illustrated in Fig. 3. All experiments in this section are conducted on the V* dataset(Wu & Xie (2023)).

### 3.1 DECOUPLING THE ZOOM-IN OPERATION: CROPPING AND UPSCALING

Previous works suggest that small objects can impair model's judgment and propose a zoom-in method (crop + upscale) to enhance perception. To isolate the effects of cropping and magnification, we uniformly upscale each image without cropping. From the left panel of Fig. 3, we observe that even when the target object size matches the size achieved by zoom-in, the model still fails.

To achieve a comprehensive analysis, we upscale dataset images by various factors and evaluate the performance of Qwen2.5-VL and InternVL3. Results in the right panel of Fig. 3 demonstrate that simply enlarging the object does not deliver stable gains; on multi-object tasks, magnification can

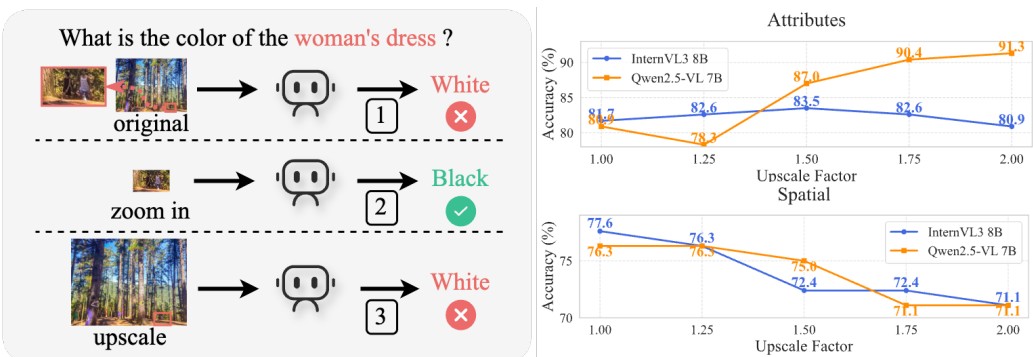

Figure 3: Left: A contradictory example comparing the inference results of zoom-in and simple resolution upscaling at the same upscale factor. Right: Performance curves showing the impact of resolution scaling on two models across two tasks—Attributes for single-object tasks and Spatial for multi-object tasks.

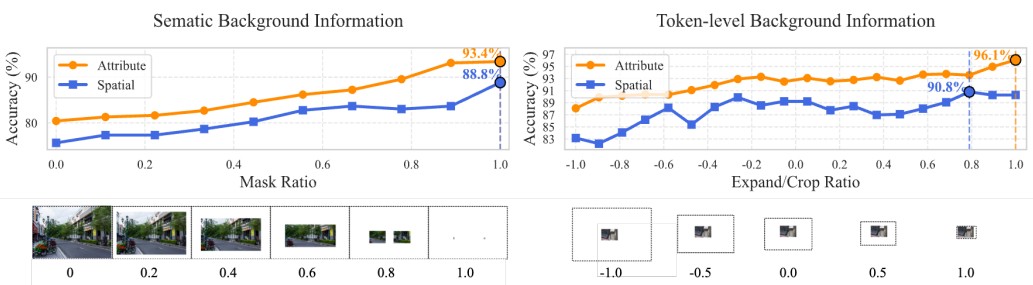

Figure 4: **Background Information Ablation Experiments.** Left: Model accuracy increases as the mask ratio of background semantic information rises. Right: Model accuracy improves as the number of background tokens decreases. Each point represents the average accuracy over 10 steps.

even hurt performance. This indicates that **zoom-in works primarily because cropping removes large amounts of irrelevant high-resolution background, not because the upscaling makes the model "sees more clearly"**. This hypothesis aligns with findings in text-only settings(Liu et al. (2025)): critical evidence can be obscured by redundant information in long texts.

### 3.2 Decoupling the crop operation: Removing semantic and token-level background information.

We further decouple the effects of cropping into two components: removing background semantics and reducing token-level redundancy, and evaluate their individual contributions.

**(1) Removing background semantics.** Using transparent masks at fixed resolution, we progressively mask non–ground-truth (non-GT) regions of each image. The mask ratio in [0, 1] denotes the fraction of background removed (1: all non-GT masked; 0: original). With 100 granularity steps, we evaluate Qwen2.5-VL at each step. As shown in Fig. 4 left, performance increases monotonically with mask ratio on both single and multi-object tasks. This demonstrates that complex background semantics significantly distract MLLMs.

**(2) Reducing token-level redundancy.** Building on experiments in (1), masked images still include large transparent areas that the vision encoder turns into redundant tokens. To investigate the effects of them, we trim these areas and, for contrast, also expand them. The results in Fig. 4 right shows that, as token-level redundancy decreases, accuracy improves; expanding transparent padding produces the opposite trend. This indicates that redundant tokens not only increase computational overhead but also interfere with MLLM attention and reasoning.

Together, these results show that **MLLMs are sensitive to both semantic background distractors and excess background tokens**. Cropping helps because it simultaneously removes semantic clutter and cuts token load.

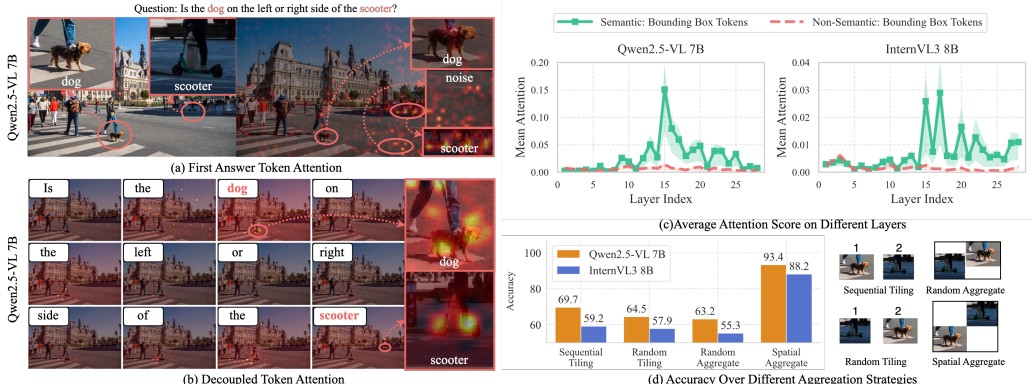

Figure 5: (a, b) Visualization of attention maps. (a): Attention map from the first generated answer token, miss some target regions and has noise. (b): Attention maps for every input question token, accurately localizing target regions based on corresponding tokens. (c): Relative attention to the bounding box areas across the layers for Qwen2.5-VL and InternVL3. (d): Accuracies of different aggregate methods, Spatial Aggregate is the best strategy.

## 3.3 DECOUPLING KEY OBJECTS: OBJECT LOCALIZATION AND LAYOUT MODELING

Having established that background information is the core bottleneck for MLLMs in high-resolution image understanding, we now focus on accurately isolating foreground content. We decouple the task into two components: precise object localization and modeling of their relative spatial layout.

**(1) Leveraging Token-level Attention for Accurate Region Proposals.** A common practice is to localize regions using attention weights from the first answer token, implicitly assuming all relevant semantics are concentrated there. This has two drawbacks (shown in Fig. 5 (a)): (a) It often overlooks parts of objects in multi-object tasks (e.g., the dog receives less attention than the scooter in the example). (b) This attention also attends to non-semantic regions, introducing noise that compromises the accuracy and completeness of key region identification. We observe that different text tokens carry different visual cues: function words disperse attention, whereas semantic words (especially nouns) focus attention on compact, informative areas (shown in Fig. 5 (b)).

Therefore, we decouple the attention of the first answer token by categorizing query text tokens into semantic and non-semantic groups. We then analyze their respective focus on image ground-truth regions by examining the cross-attention maps.

Specifically, for each text token $t_i$, we first compute its attention over the entire image, represented as a sequence of all image patch tokens $\mathcal{P} = \{p_1, p_2, \ldots, p_K\}$, where $K$ is the total number of patches. The attention weight distribution of $t_i$ across the full image at layer $l$ is computed using the standard scaled dot-product attention:

$$A_i^{(l)} = \text{softmax}\left(\frac{t_i^{(l)} \cdot \mathcal{P}^\top}{\sqrt{d_k}}\right), \tag{1}$$

where the $t_i^{(l)}$ is the hidden states at layer $l$, and $d_k$ is the attention head dimension.

Given the ground-truth region $R$, which corresponds to a subset of spatially contiguous patches $\{p_j \mid j \in \text{Idx}(R)\} \subset \mathcal{P}$, we extract the attention scores over these patches:

$$A(t_i, R)^{(l)} = \left\{A(t_i, p_j)^{(l)} \mid p_j \in R\right\}, \tag{2}$$

where the $p_j^{(l)}$ is the hidden states at layer $l$. The average attention score of semantic text tokens $T_{\text{sem}} = \{t_1, t_2, \ldots, t_M\}$ on the GT region is then defined as:

$$\bar{A}_{T_{\text{sem}}, R}^{(l)} = \frac{1}{|T_{\text{sem}}| \cdot |R|} \sum_{t_i \in T_{\text{sem}}} \sum_{p_j \in R} A(t_i, p_j)^{(l)}, \tag{3}$$

where $|T_{\text{sem}}|$ is the number of semantic tokens and $|R|$ is the number of image patch tokens within the GT region. This metric quantifies how well semantic words attend to relevant visual content. Non-semantic token sequences are evaluated analogously.

We compute the average $\bar{A}$ scores for all image-text pairs in the dataset and plot their trends across layers in Fig. 5 (c).Across the dataset, semantic tokens exhibit substantially higher attention to GT regions than non-semantic tokens, confirming that **token-level attention decoupling yields more accurate region proposals**. Furthermore, mid layers provide the strongest signal, likely because later layers prioritize semantic integration and reasoning over fine-grained visual detail.

**(2) Modeling Spatial Layouts for Regions Representation.** After acquiring multiple target regions through decoupled attention, how should they be fed back to the MLLM? We evaluate four strategies on the Spatial subset of V*: Sequence Tiling: tiling GT regions in scan order, top-left to bottom-right. Random Tiling: same tiles, shuffled order. Spatial Aggregate: recomposing GT regions while preserving relative positions. Random Aggregate: recomposing then shuffling positions. Across both Qwen2.5-VL and Intern3VL, preserving relative spatial layout (Spatial Aggregate) markedly improves performance (Fig. 5 (d)), **underscoring the importance of spatial layout modeling when aggregating target region.** We detail the layout-preserving aggregation in §4.2.

# 4 HiDe: Hierarchical Decoupling for Precise, Compact, and Structured Visual Representation

Based on our analysis in §3, we propose **Hierarchical Decoupling Framework (HiDe)**, a novel framework designed to enhance MLLM performance on high-resolution, fine-grained visual tasks, as illustrated in Fig. 6. HiDe addresses the challenge through two key contributions. First, HiDe performs **Token-wise Attention Decoupling (TAD)**, a process that establishes a correspondence between semantic key information and their visual regions via internal attention. This stage purifies the resulting attention maps by subtracting a pre-computed noise prior, thereby isolating the true signal from widespread, non-discriminative activations. Subsequently, we employ **Layout-Preserving Decoupling (LPD)**. This mechanism transforms the purified attention signals into concrete bounding boxes and then physically decouples these localized regions from the background canvas. It reconstructs them into a new, compact image, a process that eliminates irrelevant visual information while crucially preserving the relative spatial configuration of the targets. The following sections detail each component.

## 4.1 Token-wise Attention Decoupling (TAD)

As established in §3, individual semantic tokens can provide precise visual cues that are often diluted in the model's aggregate attention. To exploit this, our framework's first stage performs signal-level decoupling. This process, which we term **Token-wise Attention Decoupling (TAD)**, is designed to isolate these fine-grained alignments and purify them from background noise for accurate localization.

The TAD process begins by extracting a set of key information $\{t_i\}$ using an "extract" prompt as shown in Fig. 6. For each key information token $t_i$, we compute its raw attention map $A_i$ over the $N$ image tokens using equation 1 and reshape it to match the feature map dimensions $H \times W$.

However, these raw attention maps are often noisy (visual sink) (Kang et al. (2025)), containing widespread, non-discriminative activations from the general context rather than the specific key information, as demonstrated in our analysis. To address this and isolate the true signal, the TAD process incorporates a crucial purification step. First, we smooth the attention map with a Gaussian kernel $G_\sigma$ to reduce high-frequency noise:

$$\tilde{A}_i = G_\sigma * A_i, \tag{4}$$

where $*$ denotes 2D convolution. Next, we subtract a background noise prior, which is estimated from the attention patterns of semantically irrelevant tokens found in the generic "search" prompt. This subtraction purifies the attention for key information token $i$:

$$M_i^{\text{purified}} = \frac{\tilde{A}_i - \min(\tilde{A}_i)}{\max(\tilde{A}_i) - \min(\tilde{A}_i)} - \mathbb{E}_{q \in \text{SearchPrompt}} \left[ \frac{\tilde{A}_q - \min(\tilde{A}_q)}{\max(\tilde{A}_q) - \min(\tilde{A}_q)} \right]. \tag{5}$$

Figure 6: **The framework of HiDe.** Pure attention maps are obtained using TAD, followed by LPD to generate compact target region image. Both the target region image and the original image are fed into the MLLM to get the correct answer.

This operation effectively removes shared, non-discriminative patterns, yielding a purified attention map $M_i^{\text{purified}}$ that highlights regions uniquely activated by the key information.

Finally, naively computing these attention weights can cause significant GPU memory overhead. We propose an efficient scheme synergized with FlashAttention (Dao et al., 2022). Our approach first performs a forward pass using FlashAttention. Then, it precisely computes the attention weights only for the $n$ text queries, leveraging CPU offloading to reduce GPU memory consumption. This strategy successfully reduces peak GPU memory usage from over 96 GB to 20 GB.

## 4.2 LAYOUT-PRESERVING DECOUPLING (LPD)

After obtaining a set of purified attention maps $\{M_i^{\text{purified}}\}$, the next step is to extract the key regions from the attention signals. This entire stage is orchestrated by our **Layout-Preserving Decoupling (LPD)** mechanism, which transforms these abstract attention signals into a concrete, compact image representation. LPD operates in two stages: (i) discretizing continuous attention maps into spatial regions, and (ii) recomposing these regions on a new canvas free of background distractors.

In the first step, LPD translates each purified attention map $M_i^{\text{purified}}$ into a set of discrete bounding boxes. This is achieved by binarizing the normalized map using a threshold $\alpha \in [0, 1]$:

$$M_i = \mathbb{I}\left( \frac{M_i^{\text{purified}} - \min(M_i^{\text{purified}})}{\max(M_i^{\text{purified}}) - \min(M_i^{\text{purified}})} > \alpha \right), \tag{6}$$

where $\mathbb{I}(\cdot)$ is the indicator function. From the binary mask $M_i$, LPD then extracts connected components $\{c_j\}$ and converts each into an axis-aligned bounding box by taking its coordinates:

$$b_j = \left( \min_{p \in c_j} x_p, \ \min_{p \in c_j} y_p, \ \max_{p \in c_j} x_p, \ \max_{p \in c_j} y_p \right). \tag{7}$$

This step provides a precise set of bounding boxes $\mathcal{B} = \bigcup_i \{b_j\}$ that spatially ground all relevant semantic concepts.

In the second step, LPD provides a spatial layout preserving method to aggregate the regions (shown in Fig. 6). As discussed in 3.3, the naive approach of simply concatenating or masking would either destroy their vital relative spatial relationships or fail to remove the token-level interference from empty background patches. Instead, LPD employs a grid-based reconstruction process, detailed in Appendix Alg. 1, to systematically eliminate all background content while perfectly preserving the original spatial relations. It compacts the image by first defining a canonical grid from the coordinates of all bounding boxes in $\mathcal{B}$, yielding sorted grid lines $S_X$ and $S_Y$. Indicator functions, $I_c(i)$ and $I_r(j)$, identify columns and rows that contain content. A pixel at original coordinates $(x, y)$ is then transformed to its new coordinates $(x', y')$ in the compact image via the mapping $T$:

$$(x', y') = T(x, y) = \left( (x - s_{x,i}) + \sum_{l=0}^{i-1} I_c(l) \cdot \Delta x_l, \ (y - s_{y,j}) + \sum_{l=0}^{j-1} I_r(l) \cdot \Delta y_l \right), \tag{8}$$

Table 1: We report answer accuracy for multiple MLLMs on three HR-VQA tasks. The best score is highlighted in **bold**, and the second is underlined. The detail experimental settings as shown in §J.

| Method | Model | Train Free | V* Attr | V* Spatial | V* Avg | HRBench4K FSP | HRBench4K FCP | HRBench4K Avg | HRBench8K FSP | HRBench8K FCP | HRBench8K Avg |
|---|---|---|---|---|---|---|---|---|---|---|---|
| - | GPT-4o | - | - | - | 66.0 | 70.0 | 48.0 | 59.0 | 62.0 | 49.0 | 55.5 |
| - | OpenAI o3 | - | - | - | **95.7** | - | - | - | - | - | - |
| - | InternVL3 8B | - | 81.7 | 78.9 | 80.6 | 82.8 | 58.8 | 70.8 | 80.0 | 59.8 | 69.9 |
| ViCrop | InternVL3 8B | ✓ | 88.7 | 75.0 | 83.3 | 88.0 | 57.0 | 72.5 | 82.8 | 54.8 | 68.8 |
| **HiDe** | InternVL3 8B | ✓ | 92.2 | **90.8** | 91.6 | 90.0 | **63.5** | 76.8 | 92.2 | **62.0** | **77.1** |
| Δ(*vs* InternVL3 8B) | - | - | +10.5 | +12.1 | +10.0 | +6.2 | +4.7 | +6.0 | +12.2 | +2.2 | +7.2 |
| - | Qwen2.5-VL 7B | - | 80.9 | 76.3 | 79.1 | 88.8 | 54.8 | 71.8 | 84.2 | 51.5 | 67.9 |
| - | Qwen2.5-VL 32B | - | 87.8 | 88.1 | 87.9 | 89.8 | 58.0 | 73.9 | 84.5 | 56.3 | 70.4 |
| ViCrop | Qwen2.5-VL 7B | ✓ | 89.6 | 71.1 | 82.2 | 90.5 | 57.5 | 74.0 | 85.5 | 53.0 | 69.3 |
| DeepEyes | Qwen2.5-VL 7B | ✗ | 91.3 | 88.2 | 90.1 | 91.3 | 59.0 | 75.1 | 86.8 | 58.5 | 72.6 |
| **HiDe** | Qwen2.5-VL 7B | ✓ | **94.8** | 88.2 | 92.1 | **95.5** | 59.5 | **77.5** | **93.5** | 57.3 | 75.4 |
| Δ (*vs* Qwen2.5-VL-7B) | - | - | +13.9 | +11.9 | +13.0 | +6.7 | +4.7 | +5.7 | +9.3 | +5.8 | +7.5 |

where $\Delta x_l = s_{x,l+1} - s_{x,l}$ and $\Delta y_l = s_{y,l+1} - s_{y,l}$. The final image $I_{\text{compact}}$ is constructed by applying this transformation to all pixels within content-bearing cells, effectively stitching them together while discarding empty regions. Then, we input the original image, the obtained compact map, and the question into the model to generate the final answer.

## 5 EXPERIMENTS

### 5.1 MAIN RESULTS

To ensure a comprehensive evaluation, we select representative systems: (1) SOTA proprietary models, including GPT-4o (OpenAI. (2024)) and OpenAI o3 (OpenAI. (2025)); and (2) widely adopted open-source models, Qwen2.5-VL (Bai et al. (2025)) and InternVL3 (Zhu et al. (2025)). In addition, we compare our method against leading approaches in high-resolution visual understanding, including DeepEyes (Zheng et al. (2025)), which is based on reinforcement learning, and ViCrop (Zhang et al. (2025a)), a training-free method that relies on cropping and zooming. For open-source methods, except for Deepeyes whose results are directly taken from its original paper, all other methods have been re-evaluated. Table 1 summarizes the performance of all these models and methods across three challenging benchmarks: V*Bench (191 samples) (Wu & Xie (2023)), HRBench4K (800 samples) and HRBench8K (800 samples) (Wang et al. (2025)). For larger datasets such as POPE (9,000 samples) (Li et al. (2023b)), MME-RealWorld-Lite (1,919 samples) and MME-RealWorld-English (23,609 samples) (Zhang et al. (2025b)), we present the results based on Qwen2.5-VL 7B in the Appendix §G. The detail experimental settings as shown in Appendix §J.

The experimental results demonstrate that our method achieves the best overall performance among all competing approaches. It exhibits strong robustness and generalization across diverse model architectures and task types. It delivers consistent improvements not only in single-object recognition tasks such as attribute prediction (Attr) and fine-grained semantic parsing (FSP), but also in more complex multi-object scenarios involving spatial reasoning (Spatial) and compositional understanding (FCP). Moreover, unlike methods that require costly fine-tuning or iterative reasoning strategies such as tree search, HiDe is training-free and provides a plug-and-play capability. We shown more experiments like low-resolution and only the compact input experiments in §I and §**??**

### 5.2 VISUAL ATTENTION REGIONS COMPARISON

To better demonstrate the accuracy of the target regions identified by our TAD method, we visualize the clean attention weight maps generated by our approach and compare them with the attention

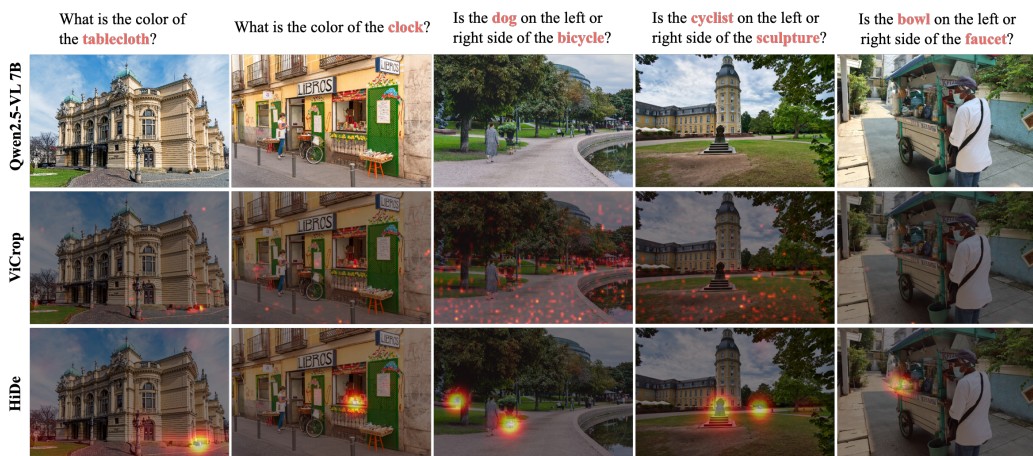

Figure 7: Comparative visualization of attention weights for HiDe and ViCrop.

Table 2: Different methods for extracting the region of interest.

| Method | Attr | Spatial | Avg |
|---|---|---|---|
| Qwen2.5-VL 7B (Base) | 80.9 | 76.3 | 79.1 |
| Base + predicted BBox | 82.6 | 82.9 | 82.7 |
| Base + Question token | 90.4 | 81.6 | 86.9 |
| Base + Decomposing semantic | 93.9 | 86.8 | 91.1 |
| Base + **TAD** | **94.8** | **88.2** | **92.1** |

Table 3: Different methods for connecting the region of interest.

| Method | Attr | Spatial | Avg |
|---|---|---|---|
| Qwen2.5-VL 7B (Base) | 80.9 | 76.3 | 79.1 |
| Base + Sequential | **94.8** | 73.7 | 86.4 |
| Base + Masking | 87.8 | 84.2 | 86.4 |
| Base + LPD w/o Compaction | 87.8 | **88.2** | 88.0 |
| Base + **LPD** | **94.8** | **88.2** | **92.1** |

weight maps corresponding to the first answer token used in ViCrop. Since TAD can generate multiple distinct attention maps, we overlay their values onto a single image to facilitate visualization. As shown in the Fig. 7, our method accurately highlights the objects of interest in both single-object and multi-object scenarios, closely aligning with the desired targets. In contrast, ViCrop's attention map appears highly scattered and disorganized in multi-object cases, failing to clearly distinguish or localize all relevant instances. This comparison underscores the effectiveness of TAD in producing more precise and semantically meaningful attention distributions.

## 5.3 ABLATION STUDY

We conduct ablation studies on V*Bench using Qwen2.5-VL 7B to evaluate two core components: TAD and LPD. All the ablation using original image and target regions images. First, for critical region extraction, we compare four progressive approaches as shown in Table 2: (1) Model-predicted BBox: bounding boxes predicted by a base model; (2) Question every token: attention from all question tokens without refinement; (3) Decomposing semantic units: attention based on extracted key information, corresponding to the initial step of TAD without purification; (4) TAD: our full token-level decoupling pipeline with attention purification. To ensure fair comparison, all detected regions are reconstructed using the LPD mechanism. Second, we evaluate four region concatenation strategies, starting from identical purified regions generated by our TAD. Results are summarized in Table 3: (1) Sequential Concatenation: regions are cropped and concatenated in sequence; (2) Masking: background is masked as transparent while preserving original layout and resolution; (3) LPD w/o Compaction: LPD preserves relative spatial layout but pads output to the original image size; (4) LPD (Ours): our method, achieving both layout preservation and compact, interference-free representation. Ablation on hyper-parameters $\sigma$ and $\alpha$ and layers is provided in Appendix §B and §H.

## 5.4 EXPERIMENT OF HIDE GENERATED BOUNDING BOXES

To quantitatively assess HiDe's bounding box accuracy, we computed IoU, precision, and recall (higher is better) against ground-truth boxes on V*Bench, comparing with ViCrop. As shown in

Table 4: Quantitative study of HiDe generated bounding boxes on V*Bench

| Metric | Method | Attr | Spatial | Avg | Method | Attr | Spatial | Avg |
|---|---|---|---|---|---|---|---|---|
| **IoU** | Internvl3 8B | 0.043 | 0.026 | 0.036 | Qwen2.5-VL 7B | 0.024 | 0.027 | 0.025 |
| **Precision** | ViCrop | 0.044 | 0.027 | 0.037 | ViCrop | 0.024 | 0.028 | 0.026 |
| **Recall** | Single box | 0.747 | 0.368 | 0.596 | Single box | 0.783 | 0.465 | 0.657 |
| **IoU** | Internvl3 8B | **0.092** | 0.072 | 0.084 | Qwen2.5-VL 7B | 0.085 | **0.096** | **0.089** |
| **Precision** | **HiDe** | **0.096** | 0.075 | **0.088** | **HiDe** | 0.087 | **0.100** | **0.093** |
| **Recall** | Multiple boxs | 0.816 | 0.806 | 0.812 | Multiple boxs | **0.931** | **0.889** | **0.914** |

Table 5: Results using only compact inputs. W means using both ori-image and compact image, while W/O means only using compact image.

| Model | Dataset | Base | ViCrop W | ViCrop W/O | HiDe W | HiDe W/O |
|---|---|---|---|---|---|---|
| Internvl3 8B | V* | 80.6 | 83.3 | 79.6 | **91.6** | 90.6 |
| Qwen2.5-VL 7B | V* | 79.1 | 82.2 | 74.9 | **92.1** | 90.1 |

Table 4, the results strongly support our claims: our token-level attention decoupling (TAD) effectively localizes query-relevant regions, enabling more accurate background removal and performance gains.

We further evaluated HiDe on V*Bench using only compact inputs. As shown in Table 5, using only the cropped image yields slightly lower performance than combining it with the original image, confirming that global context remains important (Zheng et al. (2025); Zhang et al. (2025a)). Nevertheless, it still achieves a substantial improvement over the base model.

## 5.5 EFFICIENCY ASSESSMENT

We evaluate all methods on the entire V*Bench benchmark (Wu & Xie (2023)). The performance comparison, summarized in Table 6, highlights the trade-offs between different approaches in terms of computational resources. While a baseline model like Qwen2.5-VL with FlashAttention (Dao et al. (2022)) operates within a reasonable memory budget, the original ViCrop (Zhang et al. (2025a)) implementation, which computes full attention weights, fails due to excessive memory requirements.

Table 6: Performance comparison on V*Bench. "OOM" denotes an Out of Memory error.

| Method | GPU Memory | Inference Time |
|---|---|---|
| Qwen2.5-VL 7B | ~20 GB | ~7 min |
| DeepEyes | ~20 GB | ~110 min |
| ViCrop | >96 GB (OOM) | N/A |
| ViCrop-o | ~20 GB | ~28 min |
| **HiDe** | ~20 GB | ~14 min |

To address this, we developed an optimized version of ViCrop (ViCrop-o) that successfully reduces GPU memory usage by 75% by selectively computing attention weights and leveraging CPU offloading. In contrast, HiDe achieves the same low memory usage as ViCrop-o but runs faster, thanks to its single forward pass and one-shot region identification, avoiding ViCrop's costly two-pass and sliding-window.

## 6 CONCLUSION

This work, through a hierarchically decoupled analysis of the "zoom in" operation, reveals that the core bottleneck in multimodal large language models for high-resolution image understanding lies not in insufficient perception. To address this, we propose HiDe, a training-free framework that precisely localizes and compactly reorganizes key visual regions. HiDe achieves SOTA performance on multiple HR-VQA benchmarks while significantly improving inference efficiency.

## 7 ETHICS STATEMENT

This work does not involve any ethical concerns. All datasets used are sourced from publicly available and open-access repositories, and the models employed are also derived from open-source projects. The research is conducted purely for academic purposes, with no commercial applications or interests involved. We adhere to standard scientific practices in data usage and model evaluation, ensuring transparency, fairness, and respect for intellectual property.

## 8 REPRODUCIBILITY STATEMENT

To ensure reproducibility, we will release all source code for our HiDe framework upon publication. Our experiments are based on publicly available models (Qwen2.5-VL, InternVL3) and benchmarks (V*Bench, HRBench4K, HRBench8K, POPE, MME-RealWorld), with detailed setup and evaluation protocols described in §J. All crucial hyperparameters, such as $\sigma$ for TAD and $\alpha$ for LPD, along with implementation details, are documented in the main paper and further analyzed in §B.

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

## A  ALGORITHM OF LPD

The pseudocode of LPD is shown in Alg. 1. Through LPD, we can compactly output the region selected by the bounding box.

---

**Algorithm 1** LPD: Layout-Preserving Decoupling

---

1: **Input:** Original image $I$, a set of bounding boxes $B = \{b_1, b_2, \ldots, b_k\}$
2: **Output:** Compact image $I_{compact}$
3: $X_{coords} \leftarrow \{0, I.width\}, Y_{coords} \leftarrow \{0, I.height\}$ ▷ Initialize with image boundaries
4: **for all** $b_j = (x_1, y_1, x_2, y_2)$ in $B$ **do**
5:     $X_{coords} \leftarrow X_{coords} \cup \{x_1, x_2\}$
6:     $Y_{coords} \leftarrow Y_{coords} \cup \{y_1, y_2\}$
7: **end for**
8: $S_X \leftarrow \text{sorted}(list(X_{coords}))$
9: $S_Y \leftarrow \text{sorted}(list(Y_{coords}))$ ▷ Create canonical grid lines
10: **function** ISCONTENTCELL$(i, j, B, S_X, S_Y)$
11:     $cell\_rect \leftarrow (S_X[i], S_Y[j], S_X[i+1], S_Y[j+1])$
12:     **for all** $b_k$ in $B$ **do**
13:         **if** Intersects$(cell\_rect, b_k)$ **then**
14:             **return** True
15:         **end if**
16:     **end for**
17:     **return** False
18: **end function**
19: $new\_W \leftarrow 0, new\_H \leftarrow 0$
20: $x\_map \leftarrow \{0 : 0\}, y\_map \leftarrow \{0 : 0\}$
21: **for** $i = 0$ **to** $len(S_X) - 2$ **do** ▷ Calculate new width and x-mapping
22:     **if** IsContentCell$(i, \text{any } j, \ldots)$ **then** ▷ If column i contains content
23:         $new\_W \leftarrow new\_W + (S_X[i+1] - S_X[i])$
24:     **end if**
25:     $x\_map[i+1] \leftarrow new\_W$
26: **end for**
27: **for** $j = 0$ **to** $len(S_Y) - 2$ **do** ▷ Calculate new height and y-mapping
28:     **if** IsContentCell$(\text{any } i, j, \ldots)$ **then** ▷ If row j contains content
29:         $new\_H \leftarrow new\_H + (S_Y[j+1] - S_Y[j])$
30:     **end if**
31:     $y\_map[j+1] \leftarrow new\_H$
32: **end for**
33: $I_{compact} \leftarrow \text{NewImage}(width = new\_W, height = new\_H)$
34: **for** $i = 0$ **to** $len(S_X) - 2$ **do**
35:     **for** $j = 0$ **to** $len(S_Y) - 2$ **do**
36:         **if** IsContentCell$(i, j, B, S_X, S_Y)$ **then**
37:             $crop\_rect \leftarrow (S_X[i], S_Y[j], S_X[i+1], S_Y[j+1])$
38:             $patch \leftarrow \text{Crop}(I, crop\_rect)$
39:             $paste\_pos \leftarrow (x\_map[i], y\_map[j])$
40:             Paste$(I_{compact}, patch, paste\_pos)$
41:         **end if**
42:     **end for**
43: **end for**
44: **return** $I_{compact}$

---

## B  ABLATION ON THE HYPER-PARAMETERS

We conduct ablation experiments on the two hyper-parameters over the full V$^*$Bench to search for the optimal values shown in Table 7 and Table 8.

Table 7: Qwen2.5-VL 7B hyper-parameters ablation experiments.

| $\sigma$ | $\alpha$ | Attr | Spatial | Avg | $\sigma$ | $\alpha$ | Attr | Spatial | Avg | $\sigma$ | $\alpha$ | Attr | Spatial | Avg |
|---|---|---|---|---|---|---|---|---|---|---|---|---|---|---|
| 1 | 0.1 | 91.3 | 81.6 | 87.4 | 2 | 0.1 | 87.8 | 80.3 | 84.8 | 3 | 0.1 | 87.8 | 76.3 | 83.2 |
| 1 | 0.2 | 93.0 | 78.9 | 87.4 | 2 | 0.2 | 91.3 | 84.2 | 88.5 | 3 | 0.2 | 90.4 | 77.6 | 85.3 |
| 1 | 0.3 | 93.9 | 86.8 | 91.1 | 2 | 0.3 | 92.2 | 84.2 | 89.0 | 3 | 0.3 | 88.7 | 84.2 | 86.9 |
| 1 | 0.4 | 93.0 | 88.2 | 91.1 | 2 | 0.4 | 93.9 | 86.8 | 91.1 | 3 | 0.4 | 91.3 | 86.8 | 89.5 |
| 1 | 0.5 | **94.8** | 88.2 | **92.1** | 2 | 0.5 | 93.0 | 84.2 | 89.5 | 3 | 0.5 | 93.0 | 84.2 | 89.5 |
| 1 | 0.6 | 93.9 | 89.5 | **92.1** | 2 | 0.6 | **94.8** | 88.2 | **92.1** | 3 | 0.6 | 92.2 | 86.8 | 90.1 |
| 1 | 0.7 | 91.3 | 88.2 | 90.1 | 2 | 0.7 | 93.0 | 85.5 | 90.1 | 3 | 0.7 | **94.8** | 88.2 | **92.1** |
| 1 | 0.8 | 93.0 | 88.2 | 91.1 | 2 | 0.8 | 93.9 | 89.5 | **92.1** | 3 | 0.8 | **94.8** | 85.5 | 91.1 |
| 1 | 0.9 | 87.8 | 84.2 | 86.4 | 2 | 0.9 | 92.2 | 88.2 | 90.6 | 3 | 0.9 | 92.2 | **92.1** | **92.1** |

Table 8: InternVL3 8B hyper-parameters ablation experiments.

| $\sigma$ | $\alpha$ | Attr | Spatial | Avg | $\sigma$ | $\alpha$ | Attr | Spatial | Avg | $\sigma$ | $\alpha$ | Attr | Spatial | Avg |
|---|---|---|---|---|---|---|---|---|---|---|---|---|---|---|
| 1 | 0.1 | 87.8 | 82.9 | 85.9 | 2 | 0.1 | 86.1 | 82.9 | 84.8 | 3 | 0.1 | 85.2 | 77.6 | 82.2 |
| 1 | 0.2 | 92.2 | 86.8 | 90.1 | 2 | 0.2 | 90.4 | 84.2 | 88.0 | 3 | 0.2 | 88.7 | 88.2 | 88.5 |
| 1 | 0.3 | 88.7 | 86.8 | 88.0 | 2 | 0.3 | 89.6 | 85.5 | 88.0 | 3 | 0.3 | 90.4 | 86.8 | 89.0 |
| 1 | 0.4 | 92.2 | 90.8 | **91.6** | 2 | 0.4 | 90.4 | 86.8 | 89.0 | 3 | 0.4 | 92.2 | 89.5 | 91.1 |
| 1 | 0.5 | 89.6 | **92.1** | 90.6 | 2 | 0.5 | 91.3 | 88.2 | 90.1 | 3 | 0.5 | 92.2 | 86.8 | 90.1 |
| 1 | 0.6 | 92.2 | 86.8 | 90.1 | 2 | 0.6 | 92.2 | 90.8 | **91.6** | 3 | 0.6 | 89.6 | 86.8 | 88.5 |
| 1 | 0.7 | 90.4 | 88.2 | 89.5 | 2 | 0.7 | **93.0** | 85.5 | 90.1 | 3 | 0.7 | 91.3 | 89.5 | 90.6 |
| 1 | 0.8 | 88.7 | 86.8 | 88.0 | 2 | 0.8 | 90.4 | 89.5 | 90.1 | 3 | 0.8 | 90.4 | 90.8 | 90.6 |
| 1 | 0.9 | 88.7 | 85.5 | 87.4 | 2 | 0.9 | 89.6 | 89.5 | 89.5 | 3 | 0.9 | 87.8 | 85.5 | 86.9 |

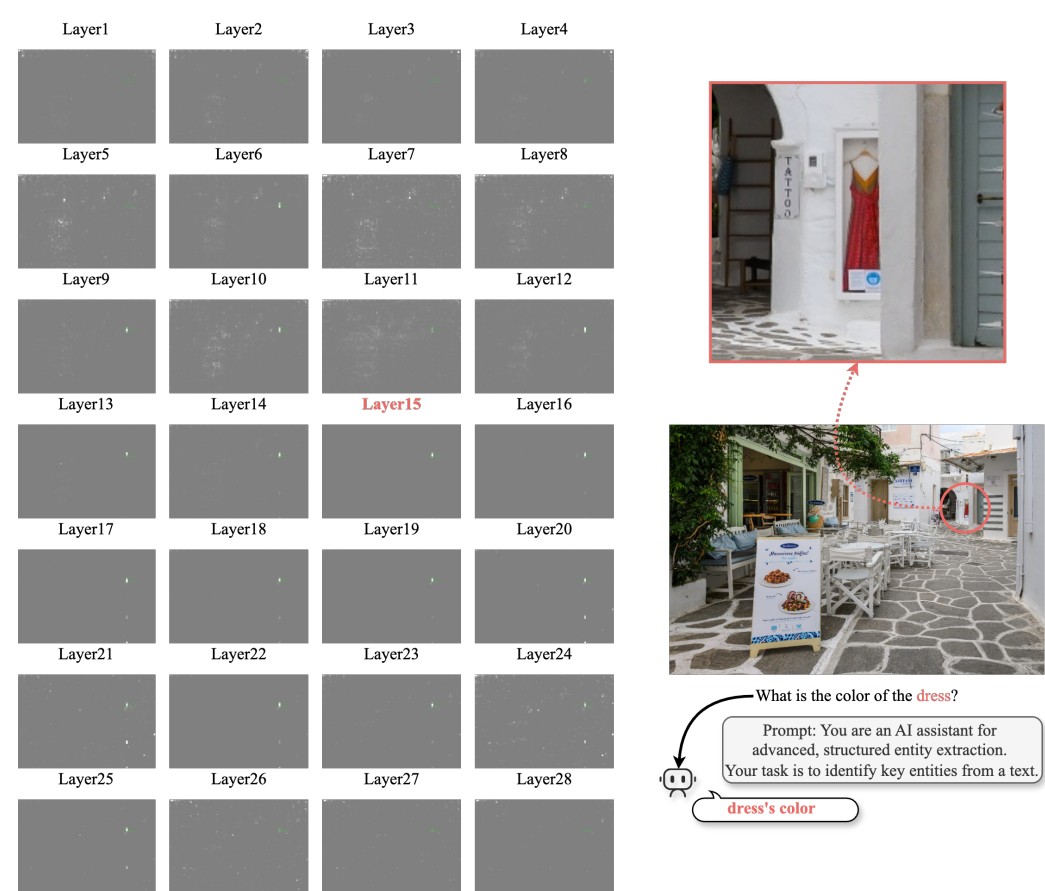

Figure 8: A sample from V*Bench focusing on the color of the dress.

## C VISUALIZATION OF ATTENTION FROM SEMANTIC UNITS AT DIFFERENT LAYERS TO THE IMAGE

We conduct a visualization study using two case examples as shown in Fig. 8 and Fig. 9 to more intuitively illustrate how semantic information from different layers attends to the image, with attention weight distributions analyzed for Qwen2.5-VL 7B.

## D TOKEN-TO-IMAGE ATTENTION MAPS ANALYSIS ON INTERNVL3-8B

Using InternVL3 8B, as shown in Fig. 10, we visualize the attention weights of each token in the generated question with respect to the image, as well as the attention weights corresponding to the first response token in a ViCrop-style manner. It can be observed that the attention weights of each decomposed token collectively cover all target regions, whereas ViCrop shows deficiencies in capturing multiple regions.

## E SOME INFERENCE CASES

We present several inference examples, including cases requiring the detection of single and multiple target regions, as shown in Fig. 11.

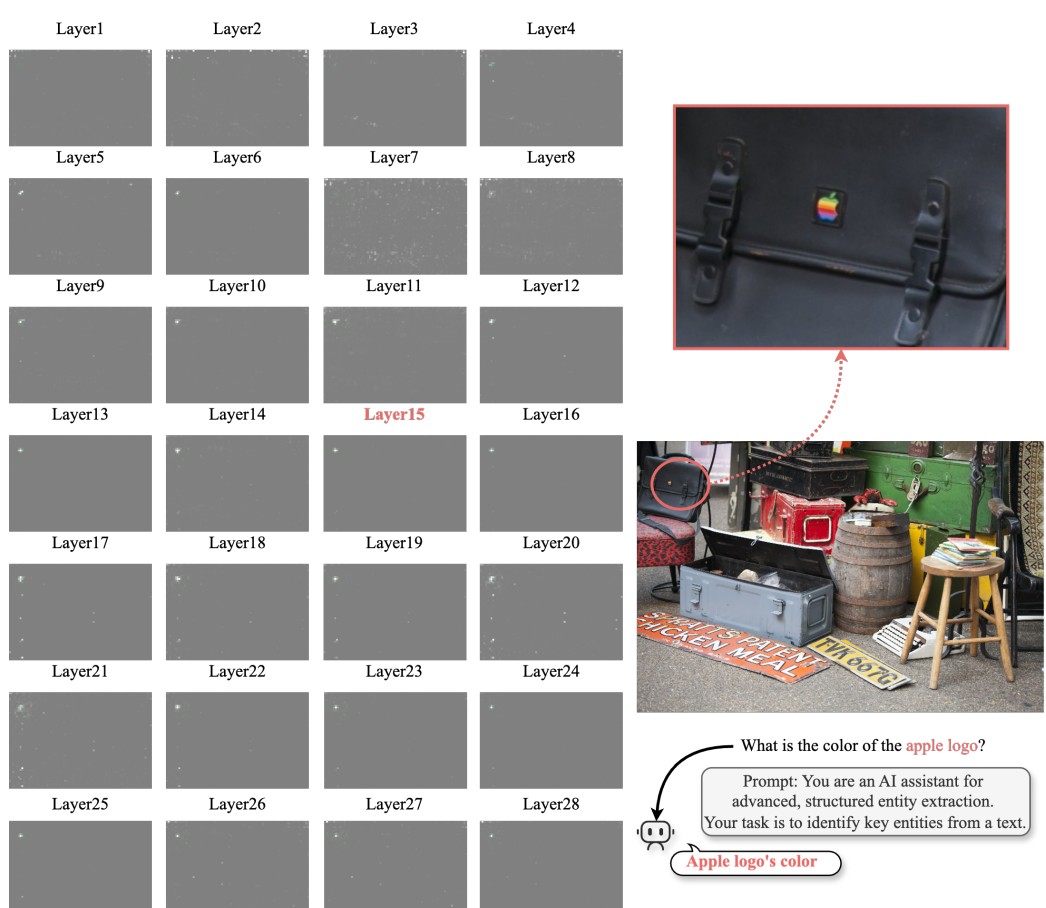

Figure 9: A sample from V*Bench focusing on the color of the apple logo.

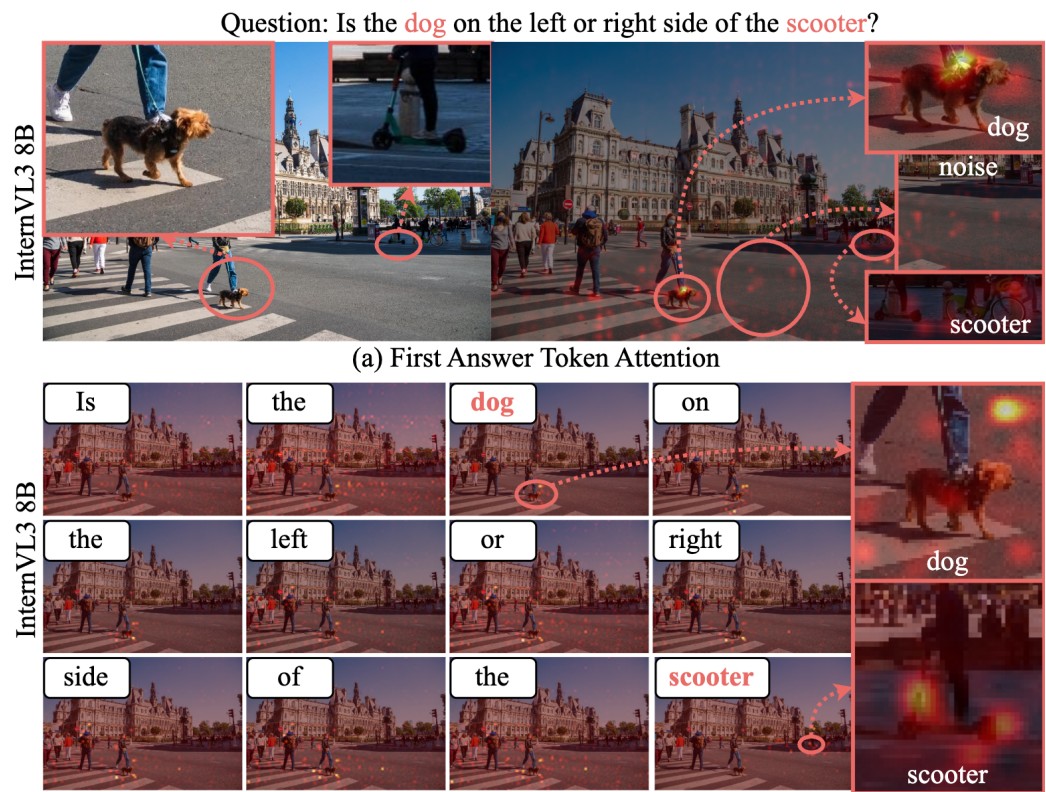

Figure 10: Token-to-image attention maps. (a): attention from the first generated answer token; (b): attention maps for individual input question tokens. Using InternVL3 8B.

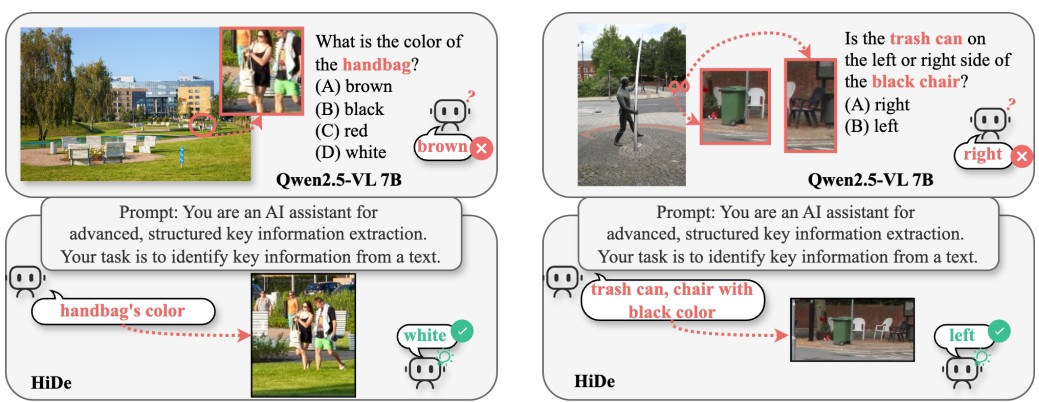

Figure 11: Left: single target region case. Right: multiple target regions case.

Figure 12: A fail case in V*Bench, requires locating the targets in the image and then determining the spatial relationship between them.

Table 9: We report answer accuracy for multiple MLLMs on other VQA tasks. The best score is highlighted in **bold**, and the second is underlined.

| Method | Model | POPE | | | | MME-RW-Lite | | | MME-RW-EN | | | DocVQA | COD10K |
|---|---|---|---|---|---|---|---|---|---|---|---|---|---|
| | | Adv | Pop | Ran | Avg | Per | Rea | Avg | Per | Rea | Avg | Acc | Acc |
| - | GPT-4o | - | - | - | - | 49.1 | **42.1** | 46.4 | 46.4 | 42.3 | 45.2 | - | - |
| - | Qwen2.5-VL 7B | 84.0 | 84.4 | 85.1 | 84.5 | 51.6 | 39.3 | 46.8 | 64.3 | 40.1 | 61.4 | 81.7 | 91.9 |
| Vicrop | Qwen2.5-VL 7B | 84.9 | 85.1 | 85.7 | 85.2 | 55.6 | 41.6 | 50.1 | 65.1 | 42.0 | 62.3 | **81.8** | 90.5 |
| **HiDe** | Qwen2.5-VL 7B | **85.1** | **85.4** | **86.3** | **85.6** | **57.8** | 42.1 | **51.7** | **66.7** | 42.9 | **63.8** | **81.8** | **92.1** |
| Δ (*vs* Qwen2.5-VL 7B) | | +1.1 | +1.0 | +1.2 | +1.1 | +6.2 | +2.8 | +4.9 | +2.4 | +2.8 | +2.4 | +0.1 | +0.2 |

## F FAILURE CASES

As shown in the Fig. 12, HiDe incorrectly localizes the target region, leading to an incorrect answer.

## G MORE RESULTS ON OTHER BENCHMARKS

POPE (9,000 samples) (Li et al. (2023b)) consists of three subsets: adversarial (Adv) (3,000 samples), popular (Pop) (3,000 samples), and random (Ran) (3,000 samples). MME-RealWorld-Lite (MME-RW-Lite) (1,919 samples) (Zhang et al. (2025b)) contains two subsets: Perception (Per) (1,169 samples) and Reasoning (Rea) (750 samples) and MME-RealWorld-English (MME-RW-EN) (23,609 samples) (Zhang et al. (2025b)) contains two subsets: Perception (Per) (20,767 samples) and Reasoning (Rea) (2,842 samples) (Zhang et al. (2025b)). We report the accuracy on each of these subsets. Otherwise, we conducted experiments on DocVQA (Mathew et al. (2020)) using ViCrop's evaluation files. We also constructed POPE-style questions using the validation set of the camouflaged object detection dataset COD10K (Fan et al. (2020)) to evaluate performance in complex scenes. Specifically, for each image, we generated two questions: one asking about objects present in the image and another asking about objects absent from the image. This resulted in a total of 4,022 questions across 2,011 images. As shown in Table 9, our method achieves improvements across different benchmark.

## H LAYER SELECTION

We selected different layers for the two base models due to their significantly different architectures, a common practice in attention-based methods (ViCrop selects a specific 22nd layer) as shown in Table 10. Although we chose the best-performing layer, we found that performance remains robust across a range of layers. As shown in the figure, performance is optimal at the best layer and slightly degrades at suboptimal layers; however, it still achieves a significant improvement over both the original model and SOTA methods.

Additionally, we conducted a new experiment demonstrating that selecting an appropriate layer requires only a small number of samples. As shown in Table 11, just a few examples are sufficient to identify a suitable layer and achieve consistent performance.

Table 10: HiDe's final performance varies depending on the selected layer.

| Model | Method | Layer | V*(Attr) | V*(Spatial) | V*(Avg) |
|---|---|---|---|---|---|
| Internvl3 8B | - | - | 81.7 | 78.9 | 80.6 |
| Internvl3 8B | ViCrop | default-22 | 88.7 | 75.0 | 83.3 |
| Internvl3 8B | HiDe | 15 | 89.6 | 86.8 | 88.5 |
| Internvl3 8B | HiDe | 16 | 91.3 | 85.5 | 89.0 |
| Internvl3 8B | HiDe | default-17 | **92.2** | **90.8** | **91.6** |
| Internvl3 8B | HiDe | 18 | 87.0 | 85.5 | 86.4 |
| Qwen2.5-VL 7B | - | - | 80.9 | 76.3 | 79.1 |
| Qwen2.5-VL 7B | ViCrop | default-22 | 89.6 | 71.1 | 82.2 |
| Qwen2.5-VL 7B | HiDe | 14 | 87.0 | 82.9 | 85.3 |
| Qwen2.5-VL 7B | HiDe | default-15 | **94.8** | **88.2** | **92.1** |
| Qwen2.5-VL 7B | HiDe | 16 | 93.0 | **88.2** | 91.1 |
| Qwen2.5-VL 7B | HiDe | 17 | 89.6 | 85.5 | 88.0 |

Table 11: The layer ultimately selected by HiDe varies depending on the number of samples chosen.

| Sample Numbers | 1 | 5 | 10 | 20 | 40 | **191** |
|---|---|---|---|---|---|---|
| Qwen2.5-VL 7B | 15 | 15 | 15 | 15 | 15 | **15** |
| Internvl3 8B | 17 | 15 | 17 | 17 | 17 | **17** |

## I    LOW-RESOLUTION EXPERIMENTS

Since our method is developed and analyzed under high-resolution scenarios, it is necessary to conduct experiments in low-resolution settings to examine whether HiDe introduces any overhead or negative bias in these simpler scenarios. To simulate low-resolution conditions, we explicitly cap the maximum number of visual tokens in the model input at $512 \times 28 \times 28$, thereby effectively limiting spatial resolution during inference. As shown in Table 12, under this setting, HiDe still maintains consistent improvements over baseline models, demonstrating its robustness even when high-resolution details are unavailable or unnecessary.

## J    EXPERIMENTAL SETTINGS

**Implementation details.** We used a computing card PPU-ZW810E with maximum memory capacity of 96 GB. We fix the temperature of model outputs to 0 to ensure deterministic responses given the same input, thereby avoiding inconsistent results caused by randomness. For InternVL3 (Zhu et al. (2025)), we set the number of image input token blocks to its maximum supported value 36. For Qwen2.5-VL (Bai et al. (2025)), we set the maximum number of image input pixels to $16384 \times 28 \times 28$ and set the minimum number of image input pixels to $256 \times 28 \times 28$, aligning with the minimum image

Table 12: Comparative experiments with a cap on the maximum number of tokens.

| Method | Model | V* | | | HRBench4K | | | DocVQA |
|---|---|---|---|---|---|---|---|---|
| | | Attr | Spatial | Avg | FSP | FCP | Avg | Acc |
| - | Qwen2.5-VL 7B | 56.5 | 64.5 | 59.7 | 59.5 | 56.3 | 57.9 | 75.8 |
| ViCrop | Qwen2.5-VL 7B | 70.4 | 64.5 | 68.1 | 66.0 | 57.0 | 61.5 | 78.2 |
| HiDe | Qwen2.5-VL 7B | **73.0** | **73.7** | **73.3** | **71.8** | **60.0** | **65.9** | **79.6** |
| Δ (*vs* Qwen2.5-VL 7B) | | +20.5 | +9.2 | +13.6 | +12.3 | +3.7 | +8.0 | +3.8 |

resolution $448 \times 448$ of InternVL3. For Qwen2.5-VL 7B, we use layer 15, set $\sigma = 3$ and $\alpha = 0.7$. For InternVL3 8B, we use layer 17, set $\sigma = 2$ and $\alpha = 0.6$.

## K    THE USE OF LARGE LANGUAGE MODELS (LLMS)

Regarding the use of large language models in the writing of this paper: we only employed a large language model to revise the sentence structures and grammar of the manuscript. After revision, the text was carefully reviewed manually to ensure that the original meaning was preserved accurately.

