# OpenReview forum: "HiDe: Rethinking The Zoom-IN method in High Resolution MLLMs via Hierarchical Decoupling"
_ICLR.cc/2026/Conference — Submitted to ICLR 2026_

### Official Review · Reviewer_m4bL · 2025-10-26

**Soundness:** 2
**Presentation:** 2
**Contribution:** 2
**Rating:** 2
**Confidence:** 4

**Summary:**

This paper studies the zoom-in operations of MLLMs for high-resolution image understanding, and propose a training-free method termed HiDe, which uses the attention distributions to select the key regions and uses a layout-preserving method to convert these regions into one compact image.

**Strengths:**

1. The proposed method is a training-free solution, which can be plugged-into most existing MLLMs without another (RL) training.

2. The effectiveness of the proposed method are shown on two popular MLLMs and three hi-res benchmarks.

**Weaknesses:**

1. The writing is over-complex, making the presentation very poor and hard to read.  This paper proposes a simple yet intuitive method to address the high-resolution image understanding problem, which applies the MLLMs' attentions o select key regions and then reorganizes them into a compact image for answering.

The authors could have described the proposed method in concise language, allowing readers to clearly understand the work and implementation. However, in the submitted manuscript, they have piled up a lot of concepts and also unreasonably "invented" many complex yet uncommon terminologies, resulting in extremely poor readability. For instance, "to address this and isolate the true signal, the TAD process incorporates a crucial PURIFICATION?? step''.

Besides, the over-complex writing also makes the paper missing key details of the proposed method, e.g., does the MLLMs need to process hi-res images twice? And how the optimal number of key regions are determined?

2. The findings are of limited interests to the community, and some arguments are in-fact very subjective. For instance, in Line 69-72, the authors argue that "scaling the whole image alone does not help MLLM decisions''. But Fig.3-left shows that scaling is beneficial at most cases, although it is not stable for multi-object tasks. The analysis of 3.1 seems lacking of enough quantitative supports.  The argument  of "the indicates that zoom-in works primarily because cropping removes large amounts of irrelevant high-resolution background, not because the upscaling makes the model “sees more clearly”." is also not convincing enough.

 Besides, other findings like "removing background semantics", "reducing token-level redundancy (i.e., the transparent padding) ",  and "semantic and non-semantic tokens (words) " are not new to the VL community.

3. The hyper-parameter selection is illy conducted. It is not appropriate to use the benchmark data to ablate the hyper-parameters. The authors seem using the subset of V* to ablate the optimal layer of attentions and image layout, i.e., Fig. 5-c d ?

**Questions:**

Q1. How the proposed method convert the selected tokens to key regions?

Q2. How to decide the optimal number of key regions for consorting a compact image? And how the model handle the cases of referring image questions based on several key regions (some are incorrect)?

Q3. What about the result of using only the reconstructed region image? And is the original image hi-res or low-res, or just following the default setting of MLLMs?

---

> ### Author Response · Authors · 2025-11-20
> **Official comment for m4bL by author 1/3**
>
> Thank you for taking the time to review our manuscript and provide your feedback. We will address your concerns point by point.
>
> **W1:  Concerns about writing style, complex terminology, and missing details.**
>
> Thank you for your feedback. We sincerely apologize that our paper caused you difficulty in reading. However, we believe that beyond the method itself, the insights gained and conclusions drawn from our analysis are also important as other reviewers recognized. Below, we provide a concise overview of our analytical reasoning and highlight its key contributions. We will also revise the manuscript to improve clarity and exposition.
>
> We systematically deconstructed the "zoom-in" paradigm and identified that its primary benefit stems from removing background interference (cropping) rather than simple upscaling. By further decoupling the crop operation, we revealed that query keywords can guide precise token-level attention. Also, preserving the original spatial layout is critical for complex reasoning. These insights motivated HiDe: a training-free, plug-and-play framework that extracts foreground objects based on query terms and reorganizes them into a compact view while strictly maintaining their geometric relationships.
>
> For the concerns about complex terminology, we will add more intuitive explanations of key concepts. For example, **“PURIFICATION”** in our paper refers to the denoising of raw attention maps, which are often noisy. We will add the explanation to help better understanding.
>
> Regarding your specific questions: **whether MLLMs need to process high resolution images twice, and how the optimal number of key regions is determined.** Our responses are integrated below:
>
> **Inference procedure**: Does the MLLM process the image twice? **Yes**. This was briefly mentioned in **lines 369–370**: “Subsequently, we feed the original image, the compact image, and the question together into the model to generate the final answer.” Our pipeline involves two stages: (1) a first forward pass to extract attention maps and generate the compact image; (2) a second forward pass where both the original image (for global context) and the compact image (for focused detail) are jointly provided to the MLLM to produce the final answer.
>
> **Number of key regions**: How is the number of key regions determined? Our method **does not use a fixed number**. Instead, it is dynamically determined by the binarization threshold α (Equation 6). Any region with a purified attention score above this threshold is selected. This allows our approach to flexibly adapt to scenes containing varying numbers of objects.
>
> In response, we will further simplify the exposition, clarify definitions of key terms, and add explanatory details at critical implementation points in the revised manuscript to improve overall readability.

---

> ### Author Response · Authors · 2025-11-20
> **Official comment for m4bL by author 2/3**
>
> **W2: Concerns about limited interest in our findings and the subjectivity of our arguments.**
>
> We appreciate your skepticism regarding the novelty and persuasiveness of our claims. Our primary contribution lies not merely in identifying background interference, but in systematically dissecting the widely used “zoom in” operation through empirical analysis and re diagnosing the core bottleneck in high resolution multimodal large language models (MLLMs).
>
> Regarding our argument on scaling **(lines 69–72 and Figure 3)**: We clarify that we do not claim scaling is entirely without benefit. As shown in **Figure 3 (right)**, scaling “fails to deliver consistent gains” on multi object tasks and can even “harm performance.” The instability of naive upsampling indicates it is not a fundamental solution; instead, cropping (background removal) is the key factor driving performance improvements. We will revise Section 3.1 to articulate this point more precisely.
>
> On the novelty of our findings: While concepts such as “background semantics” or “semantic tokens” have appeared in prior work, our novelty stems from two aspects: **(1) challenging the prevailing “small object perception” narrative through structured experiments, and (2) empirically quantifying how and to what extent these factors degrade model performance.** We are encouraged that other reviewers also recognized this value. We will strengthen the introduction to more clearly frame our contribution around this systematic analytical perspective.
>
> **W3: Concerns regarding hyperparameter selection.**
>
> Thank you for pointing out the concern regarding hyperparameter selection. We would like to clarify that our hyperparameters were not meticulously tuned on the test set. Instead, we fixed the hyperparameters after analysis on a single dataset, and these same hyperparameters generalize effectively across multiple datasets. As shown in Table 1 of the main paper, HiDe achieves new state of the art results on these completely unseen datasets (HRBench4K and HRBench8K), even outperforming training based methods. This provides strong evidence that our method and its hyperparameters generalize well and are not overfit to V*Bench.
>
> Furthermore, as shown in the appendix and the table below, we also provide additional results on larger and more diverse benchmarks such as POPE and MME. HiDe consistently delivers performance gains using the same hyperparameters, further confirming its effectiveness and robustness across different tasks.
>
> | Method | Model           | POPE(Adv) | POPE(Pop) | POPE(Ran) | POPE(Avg) | MME-RW-Lite(Per) | MME-RW-Lite(Rea) | MME-RW-Lite(Avg) | MME-RW-EN(Per) | MME-RW-EN(Rea) | MME-RW-EN(Avg) | DocVQA(ACC) |
> |--------|------------------|-----------|-----------|-----------|-----------|-------------------|-------------------|-------------------|-----------------|-----------------|-----------------|--------------|
> | -      | Qwen2.5-VL 7B   | 84.0      | 84.4      | 85.1      | 84.5      | 51.6              | 39.3              | 46.8              | 64.3            | 40.1            | 61.4            | 81.7         |
> | ViCrop | Qwen2.5-VL 7B   | 84.9      | 85.1      | 85.7      | 85.2      | 55.6              | 41.6              | 50.1              | 65.1            | 42.0            | 62.3            |  **81.8**         |
> | HiDe   | Qwen2.5-VL 7B   | **85.1**      |  **85.4**      |  **86.3**      |  **85.6**      |  **57.8**              |  **42.1**              |  **51.7**              |  **66.7**            |  **42.9**            |  **63.8**            |  **81.8**         |
>
> Finally, we would like to clarify that **Figure 5(c)** is **not a hyperparameter tuning plot but an analytical visualization**. Its purpose is to illustrate how attention scores from semantic tokens toward ground truth regions evolve across all network layers, helping us understand why certain layers are more effective. This analysis informed our design choices but was not part of a direct hyperparameter tuning procedure. We will revise the figure caption and main text to make this distinction clearer.

---

> ### Author Response · Authors · 2025-11-20
> **Official comment for m4bL by author 3/3**
>
> **Q1: How does the proposed method convert selected tokens into key regions?**
>
> Thank you for this question. The process is described in detail in **Section 4.2** and can be summarized as follows:
>
> First, for each key semantic token, we obtain its purified attention map.
>
> Next, we binarize this map using threshold α **(Equation 6)** to produce a binary mask, and identify all connected components within this mask.
>
> Finally, we generate tight bounding boxes for each connected component **(Equation 7)**; the union of all these bounding boxes forms our set of key regions.
>
> **Q2: How to decide the optimal number of key regions for consorting a compact image? And how the model handle the cases of referring image questions based on several key regions (some are incorrect)?**
>
> Thanks for your question. As noted in our response to **Weakness 1**, **the number of regions is not fixed** but dynamically determined by the threshold, providing inherent flexibility.
>
> Regarding robustness to erroneous regions, our method exhibits resilience for two reasons:
>
> First, our LPD mechanism preserves the relative spatial layout of all selected regions. Even if spurious regions are included, as long as the primary objects are present and their relative positions are accurate, the MLLM can still perform correct reasoning.
>
> Second, as stated in lines **369–370** of the main text, during final inference we feed both the compact image and the original high resolution image into the model simultaneously. The original image provides global context, enabling the model to cross reference information and potentially disregard noisy regions in the compact view.
>
> **Q3: What are the results when using only the reconstructed region image? Is the original image high resolution, low resolution, or simply the MLLM’s default input?**
>
> Thank you for this question. As you suggested, we have conducted experiments using only the compact image:
> | Model           | Dataset     | Base  | ViCrop W | ViCrop W/O | **HiDe W** | HiDe W/O |
> |-----------------|-------------|-------|----------|------------|--------|----------|
> | Internvl3 8B    | V*          | 80.6  | 83.3     | 79.6       | **91.6**   | 90.6     |
> | Qwen2.5-VL 7B   | V*          | 79.1  | 82.2     | 74.9       | **92.1**   | 90.1     |
>
>
> As shown in the table, using only the cropped image yields slightly lower performance compared to combining it with the original image. This is because global context remains important, a point also noted in [1,2]. Nevertheless, it still achieves a substantial improvement over the base model.
>
> Regarding the resolution issue you raised: the **“original image”** always refers to the **native high resolution image** provided by the benchmark dataset. Our method first analyzes this high resolution image to extract key regions, and then feeds both the original image and the compact image into the model during the final inference stage.
>
> In total, we thank you for your advice. We hope the above responses and additional experimental results have adequately addressed your concerns and further highlight the contribution of our work. If the reply helps resolve your concerns, we kindly ask you to consider raising your rating.
>
> [1]. MLLMs Know Where to Look: Training-free Perception of Small Visual Details with Multimodal LLMs. ICLR2025
>
> [2]. DeepEyes: Incentivizing" Thinking with Images" via Reinforcement Learning. ArXiv

---

### Official Review · Reviewer_46Tk · 2025-10-26

**Soundness:** 3
**Presentation:** 3
**Contribution:** 2
**Rating:** 6
**Confidence:** 3

**Summary:**

This work addresses the MLLMs' ability to process high-resolution images. The authors first argue that the primary issue is not the recognition of small objects but rather the interference from complex backgrounds. To solve this, the authors propose a training-free method that first uses Token-wise Attention Decoupling to identify key visual information based on the question and then employs Layout-Preserving Decoupling to isolate these regions from the background. This approach achieves promising results on multiple benchmarks while using 75% less memory than previous training-free methods.

**Strengths:**

1. The paper is well-written. The author conducted a detailed analysis to illustrate the problem.

2. The proposed method simply yet effectively enhances the model's capability in processing high-resolution images.

**Weaknesses:**

1. The experiments are conducted on a limited set of benchmark datasets. Many benchmarks, such as InfoVQA, TextVQA, and DocVQA, also primarily consist of high-resolution images. Is the proposed method effective on these benchmarks as well?

2. The study lacks validation of the method's performance across models of different sizes. It is unclear to what extent the method's efficacy depends on the model's inherent capabilities, and how it would perform on smaller or larger models.

3. LPD generates a series of bounding boxes, but there is a lack of quantitative metrics to verify whether these bounding boxes actually correspond to the locations of key information. It would be better if the authors could validate whether these bounding boxes indeed focus on key information through some quantitative ablation studies.

**Questions:**

See the weaknesses.

---

> ### Author Response · Authors · 2025-11-20
> **Official comment for 46Tk by author 1/2**
>
> We sincerely thank you for your valuable time and for your thoughtful, constructive feedback on our work. We are especially greatful for your acknowledgment of the good writting,thorough analysis and effective method of our paper.
>
> In response to your comments, we address each of your points below and will incorporate these clarifications into the revised manuscript.
>
> **W1: Need for results on additional benchmark datasets**
>
> Thank you for your suggestion. In addition to the two HR datasets, we also conducted experiments on larger real world datasets, as reported in Table 7 of the appendix.
>
> Furthermore, we performed additional experiments on DocVQA as you suggested, with results shown in the table below, our method consistently yields moderate improvements across these benchmarks.
>
> Since DocVQA is primarily focused on OCR based recognition, which differs from our main task, the performance gain is less pronounced compared to that on V*Bench. Nevertheless, our method does not degrade the base model’s generalization capability, confirming that HiDe functions as an incremental, plug and play module.
>
> | Method | Model           | POPE(Adv) | POPE(Pop) | POPE(Ran) | POPE(Avg) | MME-RW-Lite(Per) | MME-RW-Lite(Rea) | MME-RW-Lite(Avg) | MME-RW-EN(Per) | MME-RW-EN(Rea) | MME-RW-EN(Avg) | DocVQA(ACC) |
> |--------|------------------|-----------|-----------|-----------|-----------|-------------------|-------------------|-------------------|-----------------|-----------------|-----------------|--------------|
> | -      | Qwen2.5-VL 7B   | 84.0      | 84.4      | 85.1      | 84.5      | 51.6              | 39.3              | 46.8              | 64.3            | 40.1            | 61.4            | 81.7         |
> | ViCrop | Qwen2.5-VL 7B   | 84.9      | 85.1      | 85.7      | 85.2      | 55.6              | 41.6              | 50.1              | 65.1            | 42.0            | 62.3            |  **81.8**         |
> | HiDe   | Qwen2.5-VL 7B   | **85.1**      |  **85.4**      |  **86.3**      |  **85.6**      |  **57.8**              |  **42.1**              |  **51.7**              |  **66.7**            |  **42.9**            |  **63.8**            |  **81.8**         |
>
> We will include these new experiments and a detailed discussion in the appendix of the final version of the paper.
>
> **W2: Lack of performance validation across models of different scales.**
>
> Thank you for your insightful suggestion. In response to your query about how HiDe performs across models of different scales, we conducted corresponding experiments on **Qwen2.5 VL 3B**, with results shown below.
>
> | Method | Base_model      | Vstar(Attr) | Vstar(Spatial) | Vstar(Avg) | HRBench4k(FSP) | HRBench4k(FCP) | HRBench4k(Avg) |
> |-|-|-|-|-|-|-|-|
> | - | Qwen2.5-VL 3B   | 80.9  | 61.8| 73.3   | 83.0  | 50.3     | 66.6|
> | ViCrop | Qwen2.5-VL 3B   | 81.7 | 65.8  | 75.4| 86.3  | 49.8   | 68.0 |
> | **HiDe**   | **Qwen2.5-VL 3B**   | **85.2**| **72.4**    | **80.1**| **87.8**   | **51.5**  | **69.6** |
> | -      | Qwen2.5-VL 7B   | 80.9  | 76.3   | 79.1 | 88.8   | 54.8 | 71.8 |
> | ViCrop | Qwen2.5-VL 7B   | 89.6  | 71.1|  82.2 | 90.5  | 57.5 | 74.0 |
> | **HiDe**   | **Qwen2.5-VL 7B**   | **94.8** | **88.2**  | **92.1**| **95.5** | **59.5**| **77.5**|
>
> Our method substantially improves the performance of the **3B model**, achieving a **+6.8%** gain on V*Bench and a **+3.0%** gain on HRBench4k. This demonstrates that the benefit from removing background distractions is fundamental and not limited to larger, more powerful models. These experiments strongly confirm that HiDe’s effectiveness is independent of model scale and offers an attractive trade off between performance and efficiency. We will update the main results table with these findings and include a detailed analysis of this point in the final version of the paper.

---

> ### Author Response · Authors · 2025-11-20
> **Official comment for 46Tk by author 2/2**
>
> **W3: Lack of quantitative metrics for LPD generated bounding boxes.**
>
> We thank you for your keen observation.
> Following your suggestion, we conducted a new quantitative analysis on **V*Bench**, a dataset that provides ground truth segmentation masks for target objects. We computed the Intersection over Union (**IoU**), **precision**, and **recall** between the bounding boxes generated by our method and the ground truth bounding boxes. We compare HiDe against ViCrop using two different models.
>
> | Method | Model     | Metric     | V*(Attr)   | V*(Spatial)| V*(Avg)    |
> |--------|------------------|------------------|--------------------|--------------------|--------------------|
> | ViCrop | InternVL3 8B    | Mean IoU   | 0.043104     | 0.026017     | 0.036305     |
> |  || Mean Precision   | 0.043527     | 0.027074     | 0.036981     |
> |  || Mean Recall| 0.747318     | 0.368051     | 0.596405     |
> | HiDe   | InternVL3 8B    | Mean IoU   | **0.091759** | 0.072205    | 0.083979    |
> |  || Mean Precision   | **0.095986** | 0.074994    | 0.087633    |
> |  || Mean Recall| 0.815504    | 0.806363    | 0.811867    |
> | ViCrop | Qwen2.5-VL 7B   | Mean IoU   | 0.023698     | 0.027273     | 0.025120     |
> |  || Mean Precision   | 0.023998     | 0.028183     | 0.025663     |
> |  || Mean Recall| 0.783042     | 0.465413     | 0.656656     |
> | HiDe   | Qwen2.5-VL 7B   | Mean IoU   | 0.084756    | **0.096278** | **0.089341** |
> |  || Mean Precision   | 0.087304    | **0.100365** | **0.092501** |
> |  || Mean Recall| **0.931040** | **0.889320** | **0.914440** |
>
> These results provide strong quantitative evidence supporting our claims. They indicate that our token level attention decoupling (TAD) is highly effective at precisely localizing semantic regions mentioned in the query. This accurate localization is precisely what enables the subsequent background removal to yield performance gains.
>
> We will include this new quantitative ablation study and accompanying discussion in the supplementary material to formally validate the effectiveness of our localization mechanism.
>
> **Finally, we sincerely thank you again for your careful and valuable review.** We hope the above responses and additional experimental results have adequately addressed your concerns and further highlight the contribution and robustness of the proposed HiDe framework. We believe these additions significantly strengthen the paper. If these new results resolve your concerns, we kindly ask you to consider raising your rating.

---

### Official Review · Reviewer_xNkP · 2025-10-31

**Soundness:** 3
**Presentation:** 3
**Contribution:** 3
**Rating:** 6
**Confidence:** 4

**Summary:**

This paper challenges the prevailing assumption in High-Resolution Multimodal Large Language Models (MLLMs) that poor performance stems mainly from limited small object perception, which historically motivated "zoom-in" methods. Through systematic analysis, the authors diagnose the core bottleneck as Complex Background Interference (CBI). To address this, the authors propose the HIDE (Hierarchical Decoupling) framework. HIDE decouples visual representation learning into two stages: (1) extracting comprehensive global context, and (2) focusing on local details while simultaneously suppressing background interference. This hierarchical approach aims to more effectively utilize high-resolution tokens by guiding the model's attention away from distracting background elements. Experiments show that HIDE significantly improves performance on high-resolution benchmarks like MME-RealWorld and achieves state-of-the-art results without relying on complicated multi-stage "zoom-in" inference procedures.

**Strengths:**

1. Novel Problem Re-diagnosis: The paper’s most significant contribution is the effective re-diagnosis of the high-resolution MLLM problem. Identifying Complex Background Interference as the primary bottleneck, rather than simply small object perception, provides a novel and compelling direction for research in this domain. This core insight is the principle that guides the entire method.

2. Elegant and Principled Solution (HIDE): The Hierarchical Decoupling strategy is logically sound. By explicitly separating the acquisition of global context from the processing of local details and incorporating mechanisms to mitigate background distraction, HIDE offers a more principled approach to high-resolution input processing than traditional brute-force cropping/zooming.

3. High Compatibility and Generality: HIDE is designed as an architectural modification or training strategy and is shown to be effectively applied to different high-resolution MLLM backbones (e.g., InternVL3, Qwen2.5-VL), demonstrating its broad applicability.

**Weaknesses:**

1. Lack of Quantification for CBI: While the analysis is persuasive, the definition and quantitative measure of Complex Background Interference (CBI) remain somewhat empirical. Providing a more formal theoretical framework or a dedicated dataset/metric to quantify CBI would more robustly support the claim that it is the universal primary cause.

2. Ambiguous Implementation Details of Decoupling: The paper needs to more clearly articulate the specific mechanisms used to "suppress background interference." Does this involve a specific loss function, architectural gating, or some form of attention mask? The method by which decoupling is mathematically enforced during training requires detailed explanation.

**Questions:**

1. Ablation Study on Hierarchical Stages: Please provide ablation study results for the two core stages of HIDE: (1) Global Context Extraction and (2) Local Decoupling/Interference Suppression. Quantify the independent contribution of each stage to the final performance gain. Is the second stage alone sufficient, or is the first stage crucial?

2. Generalization to Lower Resolution/Simple Backgrounds: How does HIDE perform on standard lower-resolution benchmarks or scenes known to have simple, non-distracting backgrounds (e.g., standard VQA datasets)? Does the hierarchical decoupling mechanism introduce any overhead or negative bias in these simpler scenarios?

---

> ### Author Response · Authors · 2025-11-20
> **Official comment for xNkP by author 1/2**
>
> We appreciate your positive feedback and are glad you recognize our "novel problem re-diagnosis" and consider HiDe an "elegant and principled solution" with high generality.
>
> We will address each of your concerns point by point and incorporate these clarifications and additional analyses into the revised manuscript.
>
> **W1: Lack of Quantification for CBI**
>
> Thank you very much for your suggestion. Regarding the CBI you mentioned, in our paper it encompasses two types of interference, both of which we have validated through quantitative experimentation. The first is semantic interference: as demonstrated in **Figure 4 left**, reducing background semantic information leads to significant performance gains. The second is token interference: as shown in **Figure 4 right**, reducing redundant visual tokens also yields substantial improvements. Beside the empirical experiments, constructing a formal theoretical framework would be our future work. Thanks for the advice again.
>
> **W2:  Ambiguity in Decoupling Implementation Details**
>
> We appreciate the opportunity to clarify these implementation details. To address your specific concern, HiDe is a strictly **training-free** framework that operates without any loss functions or parameter updates. Instead, the **"background interference elimination"** is achieved through a deterministic, two-step inference pipeline. First, via **Token-wise Attention Decoupling (TAD)**, we mathematically subtract a statistical "noise prior" from the raw attention maps to silence non-informative background signals. Subsequently, we employ **Layout-Preserving Decoupling (LPD)** to convert these purified maps into bounding boxes and reconstruct a compact image view. This process physically excludes irrelevant visual tokens while retaining the original geometric context, ensuring the model focuses only on the essential semantic regions.
>
> **Q1: Ablation study of individual components**
>
> We appreciate the opportunity to clarify the independent value of each module. To better illustrate the step-by-step gains, we summarize the progression of improvements from our ablation studies (Tables 2 & 3) below:
>
> TAD (Signal Isolation): Average accuracy goes from 79.1% (Baseline) to 86.9% (Simple Attention) and finally to **92.1%** (Our Purified TAD, **+13.0 gain**)
>
> LPD (Spatial Preservation): Accuracy goes from 86.4% (Sequential Concatenation, spatial tasks) and 86.4% (Simple Masking) to **92.1%** (Our LPD, **+5.7 gain**).
>
> These comparisons highlight the distinct roles of each stage. It is natural that TAD provides the largest initial jump by finding the "signal" within the background noise. However, the comparison in the second step shows that simply finding the object is not enough; preserving the geometric layout via LPD is critical for complex reasoning, preventing the performance drop observed in methods that destroy spatial context.

---

> ### Author Response · Authors · 2025-11-20
> **Official comment for xNkP by author 2/2**
>
> **Q2: Generalization under low-resolution or simple-background settings**
>
> Thank you for your good question. In **Appendix G and Table 7** of our paper, we reported HiDe’s performance on a broader set of benchmarks with diverse background complexities, including POPE and MME-RealWorld.
>
> Furthermore, we conducted additional experiments on the **DocVQA** benchmark. The consolidated results are as follows:
>
> | Method | Model           | POPE(Adv) | POPE(Pop) | POPE(Ran) | POPE(Avg) | MME-RW-Lite(Per) | MME-RW-Lite(Rea) | MME-RW-Lite(Avg) | MME-RW-EN(Per) | MME-RW-EN(Rea) | MME-RW-EN(Avg) | DocVQA(ACC) |
> |--------|------------------|-----------|-----------|-----------|-----------|-------------------|-------------------|-------------------|-----------------|-----------------|-----------------|--------------|
> | -      | Qwen2.5-VL 7B   | 84.0      | 84.4      | 85.1      | 84.5      | 51.6              | 39.3              | 46.8              | 64.3            | 40.1            | 61.4            | 81.7         |
> | ViCrop | Qwen2.5-VL 7B   | 84.9      | 85.1      | 85.7      | 85.2      | 55.6              | 41.6              | 50.1              | 65.1            | 42.0            | 62.3            |  **81.8**         |
> | HiDe   | Qwen2.5-VL 7B   | **85.1**      |  **85.4**      |  **86.3**      |  **85.6**      |  **57.8**              |  **42.1**              |  **51.7**              |  **66.7**            |  **42.9**            |  **63.8**            |  **81.8**         |
>
> In addition, to simulate low-resolution scenarios, we explicitly constrained the maximum number of visual tokens in the model input to **512 × 28 × 28**, effectively limiting spatial resolution during inference. Under this setting, HiDe still maintains consistent improvements over baseline models, demonstrating its robustness even when high-resolution details are unavailable or unnecessary.
>
> | Method | Base_model      | V*(Attr) |  V*(Spatial) |  V*(Avg) | HRBench4k(FSP) | HRBench4k(FCP) | HRBench4k(Avg) | DocVQA(ACC) |
> |--------|------------------|-------------|----------------|------------|----------------|----------------|----------------|--------------|
> | -      | Qwen2.5-VL 7B   | 56.5        | 64.5           | 59.7       | 59.5           | 56.3           | 57.9           | 75.8         |
> | ViCrop | Qwen2.5-VL 7B   | 70.4        | 64.5           | 68.1       | 66.0           | 57.0           | 61.5           | 78.2         |
> | HiDe   | Qwen2.5-VL 7B   | **73.0**        | **73.7**           | **73.3**       | **71.8**           | **60.0**           | **65.9**           | **79.6**         |
>
>
> **We sincerely thank the you for your valuable time and expert feedback, which have significantly strengthened the quality of our work.** We believe that incorporating the above clarifications, additional analyses, and new experimental results will more thoroughly address the your concerns. If these clarifications help resolve your questions, we kindly ask you to consider raising your rating.

---

### Official Review · Reviewer_tbYa · 2025-11-01

**Soundness:** 2
**Presentation:** 3
**Contribution:** 2
**Rating:** 4
**Confidence:** 4

**Summary:**

This paper addresses the challenge of high-resolution image understanding in Multimodal Large Language Models (MLLMs). Through systematic analysis, the authors challenge the conventional wisdom that MLLMs struggle with small objects, instead identifying background interference as the primary bottleneck. They propose HiDe (Hierarchical Decoupling), a training-free framework comprising Token-wise Attention Decoupling (TAD) and Layout-Preserving Decoupling (LPD) that achieves state-of-the-art performance on multiple benchmarks while reducing memory usage by 75%.

**Strengths:**

- The paper's primary contribution—the "rethinking" from the title—is a significant one. The methodical analysis in Section 3, which isolates the "crop" operation as the key component of "zoom-in" and empirically proves that background removal (both semantic and token-level) is the dominant factor for performance gain, is clear, convincing, and a valuable insight for the community.

-  The performance gains are substantial and impressive. A training-free method achieving a +13 point average gain on V*Bench (Table 1, Qwen2.5-VL 7B) is a major improvement. The fact that this method allows a 7B model to outperform its 32B counterpart and even SOTA RL-trained methods (DeepEyes) underscores its effectiveness.

- The memory reduction from 96GB to 20GB through clever CPU offloading and selective attention computation makes this method practically deployable, addressing a critical limitation of previous approaches like ViCrop.

**Weaknesses:**

- While memory usage is reduced, the paper doesn't thoroughly analyze the computational cost of the additional forward passes and attention computations. The 3x inference time compared to baseline (7 min → 20 min) is concerning for practical deployment.

- The decoupling analysis focuses primarily on object-centric tasks. It's unclear how well these insights generalize to other high-resolution understanding tasks like dense prediction or scene understanding.

-  While the paper claims improvements on multi-object tasks, the spatial aggregation strategy might struggle with complex scenes containing many overlapping objects.

- While technically training-free, the method requires careful hyperparameter tuning (σ, α, layer selection) that appears to be dataset and model-specific. This tuning process effectively serves as a form of optimization that undermines the "plug-and-play" claim.

- V*Bench contains only 191 samples, raising concerns about statistical significance.The hyperparameters seem specifically tuned for these benchmarks

- Layer selection criteria aren't clearly explained - why layer 15 for Qwen and layer 17 for InternVL?

**Questions:**

- How sensitive is HiDe to the quality of this keyword extraction? What if the prompt is complex and the keywords are not obvious nouns?
- How does the method perform on images where foreground and background are not clearly separable?
- Can the noise prior be learned or adapted rather than using fixed "search" prompt patterns?
- What is the failure mode when the number of objects exceeds the method's capacity?

---

> ### Author Response · Authors · 2025-11-20
> **Official comment for tbYa by author 1/3**
>
> Thank you for recognizing the value of our rethinking of the "zoom-in" paradigm and the identified background bottleneck. We also appreciate your positive remarks on our performance gains and reduced memory usage.
>
> We will address each of your concerns point by point and incorporate these clarifications and additional analyses into the revised manuscript.
>
> **W1: Computational cost and inference time**
>
> We thank you for highlighting this important issue. To enhance model capabilities on high-resolution VQA tasks, some increase in computational overhead is inevitable, as shown in [1,2]. Nevertheless, our method incurs substantially lower computational cost compared to both existing training-free and training-based approaches, and obtains the most substantial performance gain in the mean tine. The results are shown in the table below.
>
> Furthermore, we have explored a new optimization strategy that can futher help reduce inference time cost. Specifically, during TAD computation, we apply the original attention mechanism only at selected layers (Qwen2.5-vl 7B is layer 15), while retaining the efficient FlashAttention implementation for all other layers. This optimization (HiDe*) achieves a **30%** reduction in V*Bench inference time, decreasing it from approximately **20 minutes** to about **14 minutes** as shown in the table below:
>
> | Method                |Model| Type           | Inference time |V* Acc|
> |-----------------------|-|----------------|----------------|-|
> | DeepEyes              |Qwen2.5-VL 7B| RL Training    | ~110 min       |90.1|
> | ViCrop                |Qwen2.5-VL 7B| Training-free  | N/A            |82.2|
> | ViCrop-o              | Qwen2.5-VL 7B|Training-free  | ~28 min        |82.2|
> | HiDe                  |Qwen2.5-VL 7B| Training-free  | ~20 min        |92.1|
> | **HiDe\* (optimized)**     | **Qwen2.5-VL 7B**|**Training-free**  | **~14 min**        |**92.1**|
>
>
> **W2, Q1, W5: Generalization and Robustness: Analysis Across Prompts, Tasks, and Benchmarks.**
>
> **For W2**: although our current benchmark is object-centric, we argue that the core principle of “eliminating background interference” is broadly applicable. For scene-level understanding tasks, the relevant “semantic keywords” may correspond to broader concepts such as “atmosphere” or “room layout.” Our TAD module subsequently focuses on regions associated with these concepts.
>
> **For Q1**: if the model fails to extract explicit keywords, it defaults to using the entire input sentence and performs token-level disentanglement (rather than keyword-based disentanglement). This form of disentanglement remains effective: compared to methods like ViCrop that rely solely on the last token, our approach enables finer-grained extraction of key targets by decoupling the model’s attention on the image at the moment each token is processed. This is evidenced by the third row of Table 2 in our ablation study (“Base + Question token”).
>
> **For W5**: we evaluated our method on HRBench using hyperparameters fixed from tuning on V*Bench. As shown in Table 1, performance consistently improves. Our strong performance on diverse datasets such as POPE and DocVQA (shown in the table below) further demonstrates that our method extends beyond simple object recognition and exhibits broader generalization capabilities.
>
> | Method | Model           | POPE (Adv) | POPE (Pop) | POPE (Ran) | POPE (Avg) | MME-RW-Lite (Per) | MME-RW-Lite (Rea) | MME-RW-Lite (Avg) | MME-RW-EN (Per) | MME-RW-EN (Rea) | MME-RW-EN (Avg) | DocVQA (ACC) |
> |--------|------------------|------------|------------|------------|------------|--------------------|--------------------|--------------------|------------------|------------------|------------------|---------------|
> | -      | Qwen2.5-vl 7B    | 84.0       | 84.4       | 85.1       | 84.5       | 51.6               | 39.3               | 46.8               | 64.3             | 40.1             | 61.4             | 81.7          |
> | ViCrop | Qwen2.5-vl 7B    | 84.9       | 85.1       | 85.7       | 85.2       | 55.6               | 41.6               | 50.1               | 65.1             | 42.0             | 62.3             | **81.8**         |
> | HiDe   | Qwen2.5-vl 7B    | **85.1**       | **85.4**       | **86.3**      | **85.6**      | **57.8**               | **42.1**              | **51.7**             | **66.7**            |**42.9**            |**63.8**             | **81.8**         |
>
> [1]. MLLMs Know Where to Look: Training-free Perception of Small Visual Details with Multimodal LLMs. ICLR2025
>
> [2]. DeepEyes: Incentivizing" Thinking with Images" via Reinforcement Learning. ArXiv

---

> ### Author Response · Authors · 2025-11-20
> **Official comment for tbYa bt author 2/3**
>
> **W3, Q2: Regarding overlapping scenes and complex scenes.**
>
> This issue involves two aspects: overlapping detected regions and object occlusion that makes it difficult for the model to distinguish individual objects.
>
> For the first case **(overlapping detected regions)** our method is based on attention mechanisms and identifies relevant regions rather than precise object instances. This design elegantly handles region overlap. When multiple objects are spatially intertwined, our TAD module naturally produces an attention map that covers the entire group of objects, treating it as a single salient region. LPD then extracts and preserves this complete region. When objects are spatially distinct, they are identified as separate regions, and LPD aggregates them while preserving their relative spatial layout. Compared to approaches that attempt to isolate only a single object within each cropped region, our method is more robust.
>
> For the second case **(objects exhibit ambiguous foreground background separation due to factors such as occlusion.)** this is fundamentally a limitation of the underlying base model. The core contribution of HiDe lies in better leveraging the signals of existing model. By making more effective use of the model’s native attention, our method identifies more accurate target regions compared to prior approaches. As shown in Figure 7 of the main paper, **the attention maps produced by our method align more precisely with the target objects**.
>
> Additionally, we constructed POPE style questions using the validation set of **COD10K** [3], a dataset for camouflaged object detection, to evaluate performance in complex scenes. Specifically, for each image, we generated two questions: one asking about an object present in the image and another about an object absent from it. This yields a total of **4,022** questions across 2,011 images. The results are as follows:
>
> Qwen2.5-VL 7B: 91.9
>
> ViCrop: 90.5
>
> HiDe: **92.1**
>
> Our method still achieves improvement on this task. This demonstrates the robustness of our method under such conditions.
>
> [3]. Camouflaged Object Detection. CVPR2020

---

> ### Author Response · Authors · 2025-11-20
> **Official comment for tbYa by author 3/3**
>
> **W4/6: Regarding hyperparameter tuning and layer selection.**
>
> We thank you for raising these critical points. We have conducted extensive analyses to ensure the robustness of our method and to fully substantiate our “plug-and-play” claim.
>
> It is correct that we tuned σ and α on VBench, as you noted. Crucially, after tuning, we **fixed these hyperparameters** and applied them **directly (without any modification)** to all other datasets.
>
> **(a) On hyperparameters (σ, α) and generalization:**
>
> As shown in main text Table 1, our method achieves state-of-the-art results on completely **unseen benchmarks such as HRBench4K and HRBench8K**, strongly indicating that the chosen hyperparameters **do not overfit and generalize well**. Moreover, ablation studies in Appendix B (Tables 5 and 6) demonstrate that HiDe remains robust across a wide range of parameter values. For instance, with Qwen2.5-VL-7B, performance remains consistently high when **α ∈ [0.4, 0.8] and σ ∈ [1, 3].** By “plug-and-play,” we mean that HiDe can be applied to new multimodal large language models without retraining, and this claim is empirically validated.
>
> **(b) On layer selection:**
>
> We chose different layers for the two base model, as their architectures are quite different, which is a common practice in attention-based methods (for example, ViCrop chose a specific layer 22). Although we selected the best performing layer, we found that performance remains robust across a range of layers. As shown, performance is best at the optimal layer and degrades somewhat at suboptimal layers; however, it still yields **substantial improvements** over both the original model and the SOTA method.
>
> | Model| Method| Layer  | Attr  | Spatial | Avg|
> |-|-|-|-|-|-|
> | Internvl3 8B  | -  | -| 81.7  | 78.9 | 80.6  |
> | Internvl3 8B  | ViCrop| default-22| 88.7  | 75.0 | 83.3  |
> | Internvl3 8B  | HiDe  | 15  | 89.6  | 86.8 | 88.5  |
> | Internvl3 8B  | HiDe  | 16  | 91.3  | 85.5 | 89.0  |
> | Internvl3 8B  | HiDe  | default-17| **92.2**  | **90.8** | **91.6**  |
> | Internvl3 8B  | HiDe  | 18  | 87.0  | 85.5 | 86.4  |
> | Qwen2.5-VL 7B | -  | -| 80.9  | 76.3 | 79.1  |
> | Qwen2.5-VL 7B | ViCrop| default-22| 89.6  | 71.1 | 82.2  |
> | Qwen2.5-VL 7B | HiDe  | 14  | 87.0  | 82.9 | 85.3  |
> | Qwen2.5-VL 7B | HiDe  | default-15| **94.8**  | **88.2** | **92.1**  |
> | Qwen2.5-VL 7B | HiDe  | 16  | 93.0  | **88.2** | 91.1  |
> | Qwen2.5-VL 7B | HiDe  | 17  | 89.6  | 85.5 | 88.0  |
>
> **Additionally**, we conducted a new experiment demonstrating that selecting an appropriate layer requires only a small number of samples. As shown in the table below, just **a few examples** are sufficient to identify a suitable layer and achieve consistent performance.
>
> ### Qwen2.5-VL 7B
>
> | Sample Numbers | Peak Layer |
> |-|-|
> | 1| 15  |
> | 5| 15 |
> | 10 | 15 |
> | 20 | 15 |
> | 40 | 15|
> | Full-191 | 15|
>
> ### InternVL3 8B
>
> | Sample Numbers | Peak Layer |
> |-|-|
> | 1| 17|
> | 5| 15|
> | 10 | 17|
> | 20 | 17|
> | 40 | 17|
> | Full-191 | 17|
>
> **Q3: Could the noise prior be learned or adapted dynamically instead of relying on a fixed “search” prompt pattern?**
>
> Thank you for this suggestion. Our current approach, which uses a fixed, generic prompt, is intentionally designed to be **training-free** and simple—core advantages that align with the central contribution of this paper. While learnable or adaptive noise priors might further improve performance, learning such priors would likely require large-scale training data, which contradicts our training-free design principle.
>
> **Q4: What is the failure mode when the number of objects exceeds the method’s capacity?**
>
> The primary challenge lies in region merging. The LPD algorithm reconstructs a compact image based on the union of all detected bounding boxes. When the prompt refers to a large number of distinct regions, the union of their bounding boxes may cover nearly the entire image, causing the reconstructed image to closely resemble the original. In such cases, the benefit of background removal diminishes, though the method does not degrade performance relative to the base model. **It simply reverts to processing the full image without harm.**
>
> **Finally, we sincerely thank you for your thoughtful and valuable review comments.**
>  We hope the above clarifications, together with our newly added experimental results, have adequately addressed your concerns and further highlighted the contribution and robustness of the proposed HiDe framework. We believe these additions significantly strengthen the paper. If these new results have resolved your questions, we kindly ask you to consider raising your score.

---

### Author Response · Authors · 2025-12-01
**Summary for AC Final Decision**

Dear AC,

Thank you for reviewing our paper. We have diligently responded to all reviewers and believe we have fully addressed their concerns through clarifications and substantial new experiments.

To aid in your final decision, we present a structured summary of the dialogue, organized by reviewer:

**Reviewer tbYa:**
Reviewer tbYa acknowledged our core contribution but raised concerns about practicality, generalization, and hyperparameter tuning.
*   **Concern:** High inference time and computational cost.
    *   **Response:** We have added a comparison with **DeepEyes** in terms of processing speed validating the relative **low computational cost of our method compared to current training-free and training methods**, and further optimized Hide’s performance, reducing its runtime from 20 minutes to 14 minutes.

*   **Concern:** Generalization and robustness to complex prompts.
    *   **Response:** Proved robustness with new tests on a **camouflaged object dataset (COD10K) and general datasets (DocVQA, POPE, MME-Realworld)** and clarified the method’s handling of complex queries.

*   **Concern:** Hyperparameter tuning undermines the "plug-and-play" claim; layer selection was unclear.
    *   **Response:** Clarified that a single set of hyperparameters generalized **without changes** across all datasets, achieving SOTA results. Showed layer selection is robust and requires a few samples.

**Reviewer xNkP:**
Reviewer xNkP appreciated our novel problem diagnosis and principled solution but requested more clarity and ablation.
*   **Concern:** Lack of quantification for "Complex Background Interference" (CBI) and ambiguous training details.
    *   **Response:** Clarified that CBI was validated in our initial quantitative (Fig. 4) and that HiDe is a strictly **training-free**, deterministic inference pipeline.

*   **Concern:** An ablation study on the two hierarchical stages was missing.
    *   **Response:** We summarized the ablation results from our main paper (Tables 2 & 3), clearly isolating the significant independent contributions of both the TAD and LPD modules.

*   **Concern:** Performance on lower-resolution or simple-background images.
    *   **Response:** Ran **new low-resolution simulation experiments** and tested on **diverse benchmarks (DocVQA, POPE, MME-Realworld)**, showing consistent improvements.

**Reviewer 46Tk:**
Reviewer 46Tk found our method simple and effective but requested broader empirical validation.

*   **Concern:** Experiments were on a limited set of benchmarks.
    *   **Response:** We have provided **new results on DocVQA** as requested, which, together with the **results in (POPE, MME-Realworld)**, demonstrate the model's generalization capability.

*   **Concern:** Lack of validation across different model scales.
    *   **Response:** We conducted **new experiments on a smaller 3B model (Qwen2.5-VL 3B)**, showing that HiDe provides substantial gains (+6.8% on V*Bench), proving its effectiveness is not limited to large-scale models.

*   **Concern:** Lack of quantitative metrics to verify the accuracy of the generated bounding boxes.
    *   **Response:** We performed a **new quantitative analysis** on V*Bench, computing **IoU, Precision, and Recall** against ground-truth masks. The results provide strong, direct evidence that our method **localizes key information** far more accurately than current SOTA method.

**Reviewer m4bL:**
Reviewer m4bL raised concerns about writing clarity, novelty, and the experimental protocol.

*   **Concern:** Findings have limited interest; hyperparameter selection was "illy conducted."
    *   **Response:** We respectfully highlighted that other reviewers found our analysis and method to be a significant contribution. We then definitively refuted the claim about hyperparameter selection by reiterating that **a single set of hyperparameters generalized to achieve SOTA results on completely unseen datasets**, which is strong evidence against overfitting.

*  **Concern:** What if only the compact input is used? What is the input resolution?
    * **Response:** We present new ablation results showing that using only the compact image still maintains strong performance; however, combining it with original image yields even better results. We also clarify that our method always processes the original full high-resolution input.

**Conclusion:**
We have taken the reviewers’ feedback seriously and addressed their concerns through new experiments that reinforce our claims of robustness, generality, and efficiency. Additionally, we have updated PDF, with all revisions highlighted in blue.

We kindly ask you to consider our responses and new evidence in your final decision.
Thank you for your time and careful consideration.

Best regards,
The Authors of Submission 9014

---

### Meta-Review · Area_Chair_Gvrp · 2025-12-29

**Summary:**

The decision to reject is primarily informed by significant concerns regarding the method's limited novelty and questionable generalization.  A major critique was the method's reliance on complex, poorly explained terminology and intricate hyperparameter tuning that undermines its "plug-and-play" claim. Furthermore, the substantial increase in computational cost and memory overhead, coupled with doubts about the method's scalability to non-object-centric tasks and dense scenes.

**Reviewer Concerns:**

The rebuttal addressed some specific technical questions by providing ablation studies on individual components and additional results. However, the fundamental concerns remain outstanding. The rebuttal did not effectively refute the critique that the method is heavily engineered for specific benchmarks.

**Reviewer Scores:**

It is highly probable that all reviewers would maintain their current evaluations. Reviewer tbYa would likely retain their marginal score because the optimized inference latency remains prohibitively high compared to baselines. Similarly, Reviewer xNkP is expected to persist in their assessment, as the provided heuristic explanation of purification and the continued absence of a formal theoretical framework for Complex Background Interference leave the method's foundational validity in question. Reviewer 46Tk would likely not raise their rating, for while the additional quantitative metrics are appreciated, they do not fully alleviate broader concerns regarding the method's robustness across diverse task types and model scales. Reviewer m4bL is anticipated to maintain their rejection, as the rebuttal fails to rectify the substantial readability issues or the perceived lack of novelty relative to established prior work.

---

### Decision · Program_Chairs · 2026-01-26

Reject